# DynaMight: estimating molecular motions with improved reconstruction from cryo-EM images

Johannes Schwab [1] ✉, Dari Kimanius [1,2], Alister Burt [1,3], Tom Dendooven [1] & Sjors H. W. Scheres [1] ✉

How to deal with continuously flexing molecules is one of the biggest outstanding challenges in single-particle analysis of proteins from cryogenic-electron microscopy (cryo-EM) images. Here, we present DynaMight, a software tool that estimates a continuous space of conformations in a cryo-EM dataset by learning three-dimensional deformations of a Gaussian pseudo-atomic model of a consensus structure for every particle image. Inversion of the learned deformations is then used to obtain an improved reconstruction of the consensus structure. We illustrate the performance of DynaMight for several experimental cryo-EM datasets. We also show how error estimates on the deformations may be obtained by independently training two variational autoencoders on half sets of the cryo-EM data, and how regularization of the three-dimensional deformations through the use of atomic models may lead to important artifacts due to model bias. DynaMight is distributed as free, open-source software, as part of RELION-5.

Structure determination of biological macromolecules by single-particle analysis of cryoegnic electron microscopy (cryo-EM) images is, at heart, a single-molecule imaging technique. Together, many images of individual complexes in a cryo-EM dataset contain information about the full extent of molecular dynamics that existed in the sample when it was plunge frozen. However, stringent low-dose imaging conditions, necessary to limit radiation damage, lead to high levels of experimental noise. Averaging over multiple individual images is thus necessary to extract detailed information about the underlying three-dimensional (3D) structures of the macromolecules. Because averaging projection images of distinct structures leads to blurring in the corresponding 3D reconstruction, image classification algorithms are often used to separate cryo-EM datasets into a user-defined number of structurally homogeneous subsets[1]. Despite their effectiveness in handling cryo-EM datasets with a discrete number of conformations, classification algorithms face challenges when continuous molecular motion is present in the sample. Therefore, continuous molecular

motions in cryo-EM datasets is often considered a nuisance, rather than a rich source of information about protein dynamics.

Manifold embedding[2] represented an early attempt to describe continuous molecular motions in cryo-EM datasets, although application of this approach has been limited to a few macromolecular complexes[3,4]. A more widely used approach to deal with continuously flexing complexes has been multi-body refinement[5]. Multi-body refinement divides complexes into independently moving rigid bodies through partial signal subtraction[6–8]. Independent image alignment and reconstruction for each of the individual bodies leads to better maps than a reconstruction of the entire complex that does not take the structural variability into account. A minimum size of the individual bodies, required for their alignment, limits the applicability of multi-body refinement to relatively large complexes. More recently, deep convolutional neural networks in the form of variational autoencoders (VAEs) have been proposed to map projection images into a continuous multi-dimensional latent space[9–11]. This mapping no longer assumes the presence of a discrete, user-defined

[1]MRC Laboratory of Molecular Biology, Cambridge, UK. [2]CZ Imaging Institute, Redwood City, CA, USA. [3]Department of Structural Biology, South San Francisco, CA, USA. ✉e-mail: schwab@mrc-lmb.cam.ac.uk; scheres@mrc-lmb.cam.ac.uk

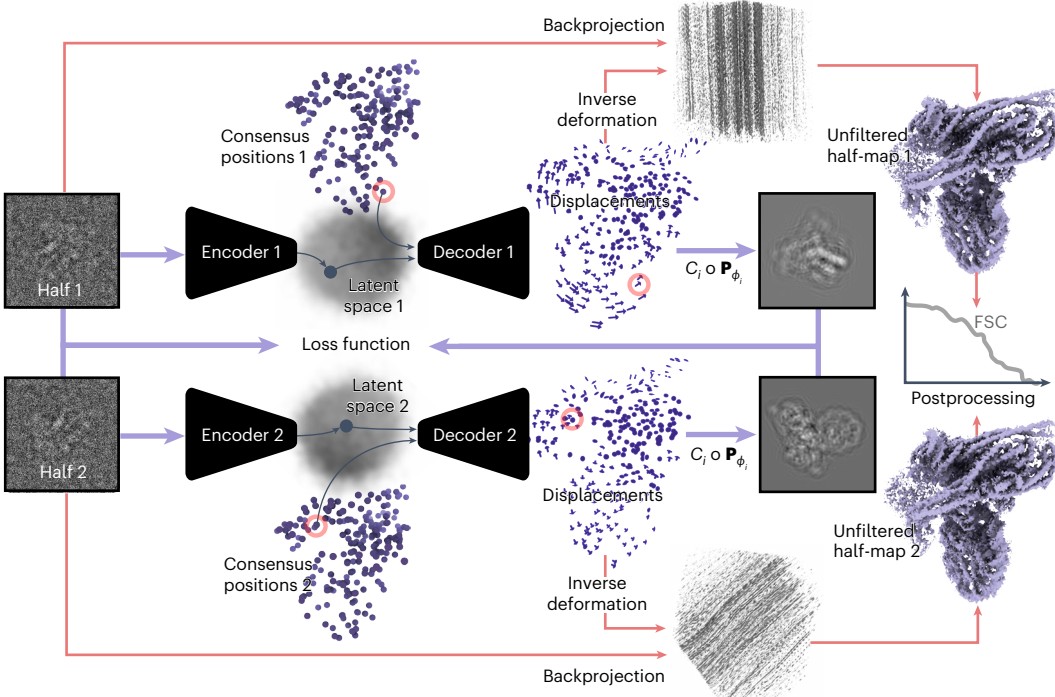

**Fig. 1 | Schematic illustration of DynaMight.** Two separate encoders take experimental images from each half set as input, and output a latent vector describing their conformational state. The decoders take the latent vectors together with the coordinates of Gaussian models for the consensus structures for each half set and generate a 3D deformation field for those Gaussians. The deformed models are then projected and compared to the experimental image in the loss function. At the end of the procedure, an approximation to the inverse deformation is used for reconstruction of an improved consensus map for each half set.

number of structures in the data. Moreover, a corresponding decoder network can be used to reconstruct 3D structures for each point in latent space, allowing the creation of movies that describe 3D protein motions by traversing latent space. These approaches have proved useful in exploring continuous molecular motions. However, in contrast to multi-body refinement, most of them do not lead to improved reconstructed densities for the moving parts.

Two methods have been proposed that aim to analyze continuous molecular motions, while also improving the reconstructed density of the underlying consensus structure. 3D flexible refinement in cryo-SPARC uses an autodecoder to learn deformations that are applied straight to the cryo-EM map[12]. A quasi-Newtonian optimization algorithm then uses the learned deformations to improve a reconstruction of the consensus structure. Alternatively, the Zernike3D approach expresses the deformation field of a cryo-EM map in a basis of 3D Zernike polynomials and uses Powell optimization to find the deformations for each individual particle image[13]. These deformations are then used in a modified algebraic reconstruction technique algorithm to obtain an improved reconstruction for the consensus structure.

In this study, we present an approach, coined DynaMight (for 'exploring protein dynamics that might improve your map'). Inspired by the approach in e2gmm[10], DynaMight uses Gaussian pseudo-atoms to model the cryo-EM density. The estimation of the conformational variability in the cryo-EM dataset is performed by a VAE, where an encoder maps individual cryo-EM images to latent space and a decoder outputs 3D deformations of the Gaussian pseudo-atoms to infer the different conformational states. We introduce a decoder architecture that takes the latent vector alongside spatial coordinates as an input and outputs actual displacements (Fig. 1). Compared to e2gmm[10], given a latent representation, the decoder directly represents the function of interest, namely a deformation field. This enables the opportunity to impose prior knowledge directly on the deformation field in the form of regularization potentials, for which we explore both benefits and pitfalls. A modified filtered backprojection algorithm, that back-projects individual particle images along curves derived from these deformations, then yields an improved density map of the consensus structure.

## Results

### Description of conformational variability

We describe the $i$th of $N_d$ particle images, $y_i$, with the following forward model:

$$y_i = \mathcal{C}_i * \mathbf{P}_{\phi_i} f(\Gamma_{z_i}(\mathbf{x})), \tag{1}$$

where $\mathcal{C}_i*$ denotes convolution with the contrast transfer function (CTF), $\mathbf{P}_{\phi_i}$ the projection of a particle that is rotated and shifted by its pose $\phi_i \in SE(3)$. We choose to represent the function $f$ by a sum of $N_g$ 3D Gaussian basis functions, or pseudo-atoms:

$$f(\mathbf{x}) \approx \hat{f}(\mathbf{x}) := \sum_{j=1}^{N_g} a_j \mathcal{G}_{s_j}(\mathbf{x} - c_j) \tag{2}$$

where $\mathcal{G}_{s_j} : \mathbb{R}^3 \to \mathbb{R}$ is $\mathcal{G}_{s_j}(\mathbf{x}) = \exp\left(\|\mathbf{x}\|^2/s_j\right)$. Here, $a_j > 0$ denote the amplitudes, $s_j > 0$ the widths and $c_j$ the central positions of the Gaussian functions.

We assume that all particle images are conformational variations of a single, consensus structure that is described by the $N_g$ 3D Gaussian basis functions and $z_i$ in equation (1) is the conformational encoding for the $i$th image. We describe the deformation of individual particles as a deviation from the consensus coordinates $\mathbf{x}$: $\Gamma(\mathbf{x}) = \mathbf{x} - \delta(\mathbf{x})$, so that:

$$
\begin{aligned}
\hat{f}(\Gamma(\mathbf{x})) &= \sum_{j=1}^{N_g} a_j \mathcal{G}_{s_j}(\Gamma(\mathbf{x}) - c_j) \\
&= \sum_{j=1}^{N_g} a_j \mathcal{G}_{s_j}(\mathbf{x} - \delta(\mathbf{x}) - c_j) \\
&\approx \sum_{j=1}^{N_g} a_j \mathcal{G}_{s_j}\left(\mathbf{x} - (c_j + \delta(c_j))\right),
\end{aligned}
\tag{3}
$$

where the last approximation assumes that the deformation field is locally constant and that the density surrounding $c_j$ moves in a similar manner. This enables us to describe the deformations as displacements of the Gaussian centers, which is a computationally tractable representation. Furthermore, the widths $s_j$ and amplitudes $a_j$ of all Gaussian pseudo-atoms are kept the same for the entire dataset. This means that DynaMight is by design constrained to only model mass-conserving heterogeneity and cannot handle nonstoichiometric mixtures. Therefore, compositional heterogeneity should be removed from the dataset by alternative approaches before running DynaMight.

### Estimation of conformational variability

For learning the deformations, we use a VAE that consist of two neural networks, namely an encoder $\mathcal{E}$ that predicts an $l$-dimensional latent representation $z_i$ per particle image, and a decoder $\mathcal{D}$ that predicts the displacement of all Gaussian pseudo-atoms in the model. The encoder is a fully connected neural network with three linear layers and rectified linear unit activation functions. The input is a (real-space) experimental image $y_i$ and the output are two vectors $(\mu_i, \sigma_i) \in \mathbb{R}^{N_l} \times \mathbb{R}^{N_l}$, which describe the mean and standard deviation used to generate a sample $z_i$ that serves as input for the decoder.

The decoder $\mathcal{D}(z_i, c_j)$ then approximates the term $c_j + \delta_j$ for each $z_i$. We define the decoder for the entire set of $N_g$ positions as:

$$\mathcal{D}(z_i, \mathbf{c^0}) = \mathbf{c^0} + \delta_\theta(z_i, \mathbf{c^0}) \qquad (4)$$

In the above, $\mathbf{c^0}$ is all the consensus positions and $\delta_\theta$ is a differentiable function, $\delta_\theta(z_i, \mathbf{c^0}) = [\delta_\theta(z_i, c_1), \ldots, \delta_\theta(z_i, c_{N_g})]$, with parameters $\theta$, that approximates $\delta$ for each position (Extended Data Fig. 1). In practice, we evaluate the decoder for each position $c_j$ and query $\delta_\theta$ with a positional encoding of $c_j$, concatenated with the latent representation $z_i$ that describes the conformation of each particle.

The output positions are used to generate a projection image $p_i$ of the deformed model in the pose of the particle, and the difference with the experimental image $\| p_i - y_i \|_\Sigma^2$ is minimized during training of the neural networks. Once trained, for a latent embedding of the whole dataset, one obtains a family of deformation fields $\mathcal{D}(z_i, \mathbf{x}) \approx \Gamma_{z_i}(\mathbf{x})$ that is defined over the entire 3D space.

### Regularization and model bias

Because of high levels of experimental noise, cryo-EM reconstruction is an ill-posed problem. Even for standard, structurally homogeneous refinement, there are many possible rotational and translational assignments for each image. When estimating conformational variability, the poses are known, but many deformed density maps may explain each experimental image equally well. Therefore, in both cases regularization is essential for robust reconstruction.

The most common form of regularization in VAEs is to constrain the distribution of latent variables to follow a Gaussian distribution, which lead to the model learning more meaningful and structured representations. The design of the decoder in Fig. 1 allows an additional form of regularization that imposes prior knowledge on its output of real-space deformation fields. A wide range of physically and biologically inspired penalties can be incorporated as priors on the deformations, also see refs. 12,14,15. Possibly a powerful source of prior information would come from an atomic model of the consensus structure, which could provide constraints on chemical bonds, maintain secondary structure elements and so on.

To explore direct regularization of the deformation fields, we tested two approaches. The first approach aims to use prior information from an atomic model that is built in the consensus map, before running DynaMight. It generates a coarse-grained Gaussian representation of the atom positions, and then minimizes changes in the distances between these Gaussians according to the bonds that exist in the atomic model:

$$\mathcal{R}(E) = \sum_{\{(i,j) : E_{ij} = 1\}} |d(c_i, c_j) - d(\mathcal{D}(c_i, z), \mathcal{D}(c_j, z))|^2, \qquad (5)$$

where $E_{i,j} = 1$ if there is a bond between the two pseudo-atoms $c_i$ and $c_j$ and $d$ denotes Euclidian distance. The deformations with this regularization scheme result in Gaussians that remain close to a coarse-grained representation of the original atomic model.

The second regularization approach uses less prior information and does not require an atomic model. Instead, Gaussians are placed randomly to fill densities in the consensus map, and connections $E$ in equation (5) are for all pairs of Gaussians that are within a distance of 1.5 times the average distance between all Gaussians and their two nearest neighbors. This regularization enforces overall smoothness in the deformations. Additional penalties that prevent Gaussians coming too close to each other, or moving too far away from other Gaussians, also exist to ensure a physically plausible distribution of Gaussians.

### Improved 3D reconstruction

We propose an algorithm that uses the estimated deformation fields $\Gamma$ to obtain an improved reconstruction of the consensus structure that incorporates information from all experimental images. To map back individual particle images to a hypothetical consensus state, one needs to estimate the inverse deformations, which represents a challenge. Whereas the inverse deformation on the displaced Gaussians is given by the negative displacement vector, that is $\Gamma^{-1}(\Gamma(c_i)) = c_i$, the inverse deformation field needs to be inferred at all Cartesian grid positions of the improved reconstruction. We train a neural network as a regression function to estimate a deformation field that coincides on the given sampling points $\Gamma(c_i)$, but can be evaluated on arbitrary positions. This network consists of an multilayer perceptron with six layers and a single additive residual connection to the original coordinates of the consensus model $\mathbf{c^0}$. Similar to the forward deformation model, the network takes the latent code $z_i$ and the deformed positions $\Gamma(c_i)$ as inputs and aims to output the original positions $c_i$. In addition to the inversion of the forward fields on the sampling points, we force the inverse field to be smooth by adding a regularization term to the loss function.

The algorithm aims to improve the reconstruction of the density $f$, using the known deformations $\Gamma$, that is we aim to find the minimizer $\hat{f}$ of the data fidelity

$$\hat{f} = \operatorname*{argmin}_f \sum_i \| \mathcal{C}_i(P_{\Gamma_i} f) - y_i \|^2. \qquad (6)$$

This minimizer can be computed using the reconstruction formula

$$\begin{aligned} \hat{f} &= \left[ \sum_i P_{\Gamma_i}^* \circ \mathcal{C}_i^2 \circ P_{\Gamma_i} \right]^{-1} \left[ \sum_i P_{\Gamma_i}^* (\mathcal{C}_i^* y_i) \right] \\ &= D^{-1} \left[ \sum_i P_{\Gamma_i}^* (\mathcal{C}_i^* y_i) \right], \end{aligned} \qquad (7)$$

to get an estimate of the unknown density $f$. Here $D$ is a matrix that depends on the estimated deformations, and $P_{\Gamma_i}^*$ is the composition of the backprojection operator and the inverse deformation corresponding to the $i$th particle (Fig. 1). For the structurally homogeneous case, $\Gamma$ is the identity operator and $D$ is diagonal in Fourier space and therefore the inverse can be computed simply by division, given that the distribution of projection directions covers the whole frequency domain and $D$ has no zeros in the diagonal. In the presence of deformations, this matrix is not diagonal anymore and would be too expensive to compute or store. We approximate equation (7) by using the filter that would correspond to the homogeneous case, without deformations. Although even in the optimal scenario of having complete data of clean projection images, this method does not yield a minimum of

functional in equation (6), it still allows to correct for the deformation to some degree. When the deformation fields are not smooth, for example when two nearby domains move in opposite directions, reconstruction with the proposed algorithm may introduce artifacts at the interface between the domains.

## Implementation details

The initial positions of the Gaussians for the VAE are obtained by approximating a map from a consensus refinement with a Gaussian model. This initial consensus map does not correspond to an actual state of the complex, but rather to a mixture of different conformations. Therefore, parts of the map will have regions of poorly defined density, and correspondingly fewer Gaussians. To overcome this limitation, we update the positions of the consensus Gaussian model throughout the estimation of the deformations, such that the positions $c_i$ may correspond to a single conformation at the end of the iterative process. We recommend using two Gaussians per residue, but a smaller number can be chosen if computational resources are limited or a low resolution estimation of the motion is required.

After initialization of the Gaussians, in the first epochs of the training of the VAE, we only optimize the global Gaussian parameters, that is their widths, amplitudes and positions. These parameters are optimized with the ADAM optimizer and a learning rate of 0.0001. After this initial warm-up phase, we start optimization of the network parameters of the VAEs, again using the ADAM optimizer with a learning rate of 0.0001. During the second phase, the parameters of the Gaussians continue to be updated. Training of the VAEs is stopped when the updates of the consensus model do not yield improvements anymore or a fixed, user-defined number of epochs are completed.

Training of the VAE is performed on two half sets, where two encoder–decoder pairs are trained independently, as illustrated in Fig. 1. This procedure yields two independent families of deformation fields, one for each half set. The approximate inverse of these deformations are then used by the deformed weighted backprojection algorithm to generate two independent maps with improved estimates for the consensus structure. These half-maps can then be used in conventional postprocessing and resolution estimation routines. As described in the 'Discussion' section, by setting aside a small validation set of images, the two independent decoders also allow an error estimation of the displacement fields.

DynaMight has been implemented in pyTorch[16], and is accessible as a separate job type from the RELION-5 graphical user interface. Because, as we will show below, the direct regularization of the deformation fields using atomic models may lead to overfitting, only the approach that enforces smoothness on the deformations, without the use of an atomic model, is exposed to the user on the graphical user interface. DynaMight uses the Napari viewer[17] to visualize the distribution of particles in latent space, as well as the corresponding deformation fields. The same viewer also allows real-time generation of densities from points in latent space, movie generation, and the selection of particle subsets in latent space.

Further implementation details are given in the Methods.

## Regularization can lead to model bias

We first analyzed the different options for regularization of the deformations on a well characterized dataset on the yeast *Saccharomyces cerevisiae* precatalytic B complex spliceosome[18] EMPIAR-(10180, ref. 19). The same data, or subsets of it, have also been analyzed using multi-body refinement[5] cryoDRGN[9], Zernike3D[13] and e2gmm[10]. To minimize computational costs and to ensure structural homogeneity[9], we used 3D classification in RELION[20] to select ~45,000 particles with reasonable density for the head region. Training of the VAEs on this subset with a box size of 320 took about 2.5 minutes per epoch on a single NVIDIA A100 GPU. This resulted in training times between 8 and 12 hours for estimating the deformations. Further estimation of

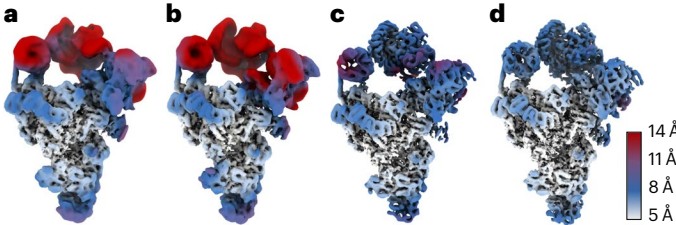

**Fig. 2 | DynaMight reconstructions of the spliceosome subset. a,** Standard RELION consensus refinement. **b,** DynaMight without regularization. **c,** DynaMight with smoothness regularization on the Gaussians. **d,** DynaMight with regularization from an atomic model. All maps are colored according to local resolution, as indicated by the color bar.

the inverse deformations took ~4 hours and reconstruction with the deformed backprojection ~3 hours on the same GPU.

Without any regularization of the deformations, estimated deformation fields displayed rapidly changing directions for neighboring Gaussians, and deformed backprojection yielded reconstructions for which the local resolution did not improve with respect to the original consensus reconstruction (Fig. 2a,b). A consensus reconstruction with better local resolutions was obtained using the regularization scheme that enforces smoothness in the deformations, but without using an atomic model (Fig. 2c). The map with the highest local resolutions was obtained using the regularization scheme that enforces distances between bonded atoms of an atomic model (Protein Data Bank (PDB) ID 5nrl) (Fig. 2d). It thus appeared that incorporation of prior knowledge from the atomic model into the VAE had been beneficial.

However, because the neural networks in our approach comprise many parameters, we were worried that there would be scope for 'Einstein-from-noise' artifacts, similar to those described for orientational assignments in single-particle analysis[21–23]. To test this, we performed two control experiments.

In the first control experiment, we replaced the atomic model of the U2 3′ domain/SF3a domain with a different protein domain of similar size (PDB 7YUY)[24]. The U2 3′ domain/SF3a showed only weak density in the consensus map, indicating large amounts of structural heterogeneity in this region. Although using the incorrect atomic model to estimate the deformation fields led to a similar improvement in local resolution compared to using the correct model (Fig. 3a,b), the reconstructed density from the deformed backprojection resembled the incorrect model, rather than the correct model (Fig. 3c and Supplementary Video 1).

In the second control experiment, we replaced the atomic model of the SF3b domain with PDB 1G88 (ref. 25). The density for the SF3b domain in the consensus map was stronger than the density for the SF3a region, indicating that this region in the spliceosome is less flexible. In this case, using the incorrect atomic model yielded a map with lower local resolutions in the SF3b region than using the correct model (Fig. 3d,e). But still, the reconstructed density from the deformed backprojection resembled the incorrect model more than the correct model (Fig. 3f and Supplementary Video 2).

These results indicate that estimation of deformation fields may lead to model bias, to the extent that reconstructed density may reproduce features of an incorrect atomic model. The scope for model bias to affect the deformed backprojection reconstruction is larger in regions of the map with higher levels of structural heterogeneity. Because it would be difficult to distinguish correct atomic models from incorrect ones, we caution against the use of this type of regularization in DynaMight. Therefore, in what follows, we only used the less informative, smoothness prior on the deformations. Using this prior, the deformations estimated by DynaMight are qualitatively similar to those observed for the same dataset using e2gmm[10] (Extended Data Fig. 2 and Supplementary Video 3). For a different set EMPIAR-(10073, on the U4/U6.U5 tri-snRNP complex[26]), using the less informative smoothness

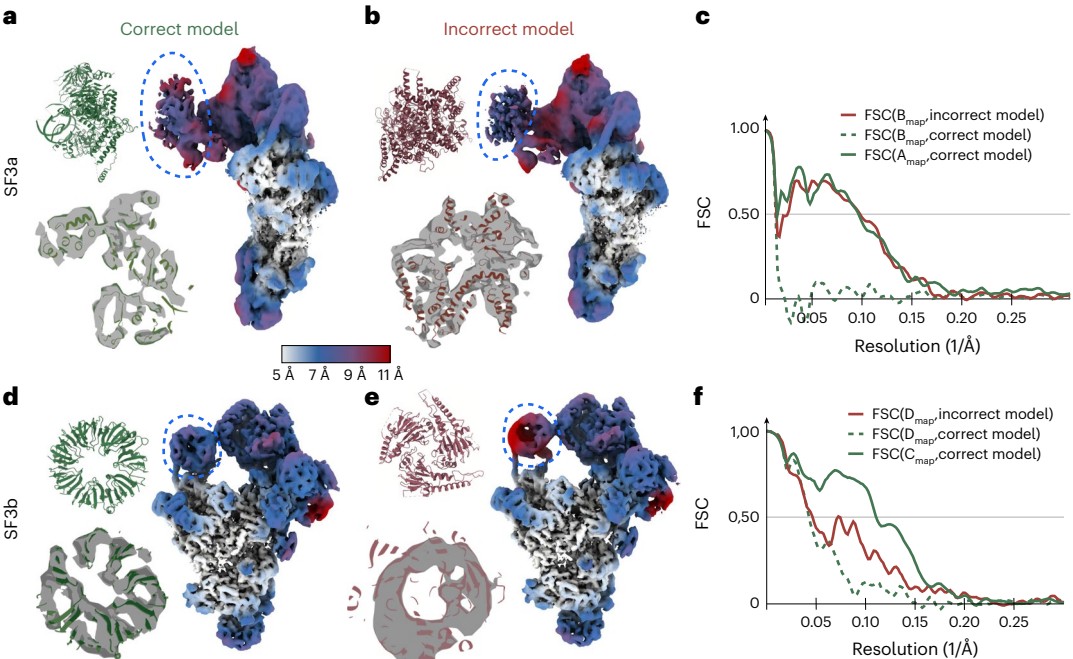

**Fig. 3 | Using incorrect atomic models in DynaMight. a**, Reconstruction after deformed backprojection using the correct atomic model for the SF3a region, colored by local resolution (right). The correct atomic model for the SF3a region is shown in green on the top left; an overlay of that model with the reconstructed density after deformed backprojection is shown on the bottom left. **b**, As in **a**, but using an incorrect atomic model for the SF3a region (shown in red). **c**, Fourier shell correlation (FSC) curves between the maps in **a** or **b**, masked around the SF3a region, and the correct (green) or incorrect (red) atomic models. **d–f**, As in **a–c**, but using different atomic models for the SF3b region: reconstruction after deformed backprojection using the correct atomic model for the SF3b region (**d**), using an incorrect atomic model for the SF3b region (**e**) and FSC curves between maps in **d** or **e**, masked around the SF3b region, and the correct or incorrect atomic models (**f**).

prior in DynaMight led to an improved reconstruction with better map features and higher local-resolution estimates than reported for 3DFlex[12] (Extended Data Fig. 3 and Supplementary Video 4), despite that 3D classification in RELION-5 selected a structurally homogeneous subset of only 86,624 particles, compared to 102,500 particles used for 3DFlex.

**DynaMight improves inner kinetochore maps**

Next, we demonstrate the usefulness of DynaMight on two cryo-EM datasets of the yeast inner kinetochore[27]. Training of the VAEs took 17 and 27 hours on an NVIDIA A100 GPU for the two respective datasets described below, with particle box sizes of 320 and 360. Estimating the inverse deformations took ~6 hours for both datasets. The deformed reconstructions took 9 and 13 hours, respectively.

The first dataset EMPIAR-(11910) comprises 100,311 particles of the monomeric constitutive centromere associated network complex bound to a CENP-A nucleosome (CCAN–CENP-A). For this dataset, we trained the half-set VAEs for 220 epochs and we used a ten-dimensional latent space. The estimated 3D deformations are distributed uniformly in latent space (Fig. 4a), without specifically clustered conformational states, suggesting that the motions in the dataset are mainly of a continuous nature. Analysis of the motions revealed that the nucleosome is rotating in different directions relative to the rest of the complex, and that these rotations coexist with the up and down bending of the Nkp1, Nkp2, CENP-Q and CENP-U subunits (arrows in Fig. 4b and Supplementary Video 5). The reconstruction from deformed backprojection improved local resolutions compared to the consensus map from standard RELION refinement, with clear improvements in the features for both protein and DNA (Fig. 4c,d and Extended Data Fig. 4).

The second data EMPIAR-(11890) comprises 108,672 particles of the complete yeast inner kinetochore complex assembled onto the CENP-A nucleosome. Training of the VAE was done for 290 epochs, and the dimensionality of the latent space was again set to ten. Again,

a continuous distribution of deformations in latent space suggests continuous structural flexibility (Fig. 5a). Analysis of the deformations revealed large relative motions between different regions of the complex (root-mean-squared deviation and additional details are given in Supplementary Table 1). Different states of the complex are depicted in Fig. 5a and Supplementary Video 6. Deformed backprojection resulted in a map with improved local resolution and protein and DNA features compared to the map from consensus refinement (Fig. 5b,c and Extended Data Fig. 4).

Because this complex, with a molecular weight of 1.5 MDa, is large enough to divide into multiple independently moving rigid bodies, we also applied multi-body refinement[5] to this dataset. We used the four bodies illustrated in Fig. 5d; body 1 (orange): CCAN$^{Topo}$, body 2 (light green): CCAN$^{Non-topo\Delta CENP-I(Body)}$, body 3 (yellow): CBF3$^{Core}$+CENP-I$^{Body}$ and body 4 (dark green): CENP-A$^{Nuc}$. The local resolutions resulting from multi-body refinement (Fig. 5d) are better than those from the deformed backprojection reconstruction of DynaMight, illustrating that there is still room for further development of the latter. Nevertheless, the DynaMight map had better protein and nucleic acid features than a map obtained for the same dataset with 3DFlex, using default parameters[12] (Extended Data Fig. 5). The DynaMight map also correlated better than the map from 3DFlex with atomic models that were built in the maps from multi-body refinement. Despite these observations, resolution estimates calculated from half-maps calculated by 3DFlex were higher than those calculated from half-maps by DynaMight. This suggests that using a single 3D deformation model in 3DFlex, rather than two separate models as done in DynaMight, could potentially result in over-estimations of local resolution.

**Discussion**

How to deal with continuous conformational heterogeneity remains a rapidly developing topic in cryo-EM single-particle analysis. As outlined

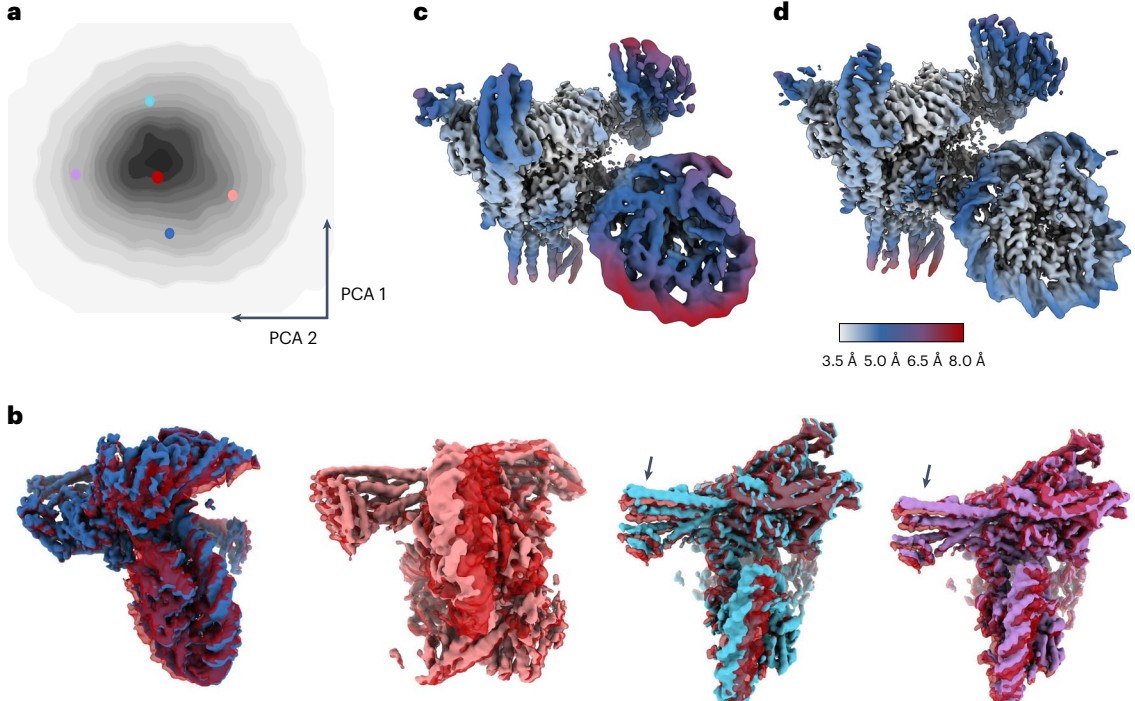

**Fig. 4 | DynaMight results for the CCAN–CENP-A complex. a**, Principal components analysis (PCA) of the conformational latent space, with colored dots indicating the positions of the five maps in **b**. (Only the latent space for one of the two half sets is shown.) **b**, Five conformational states of the complex. One state, in red, is shown in all four panels. The colors of the five maps are the same as the colors of their corresponding dots in **a**. **c**, Reconstructions from standard RELION consensus refinement. **d**, The improved reconstruction using DynaMight. The maps in **c** and **d** are colored according to local resolution, as indicated by the color bar.

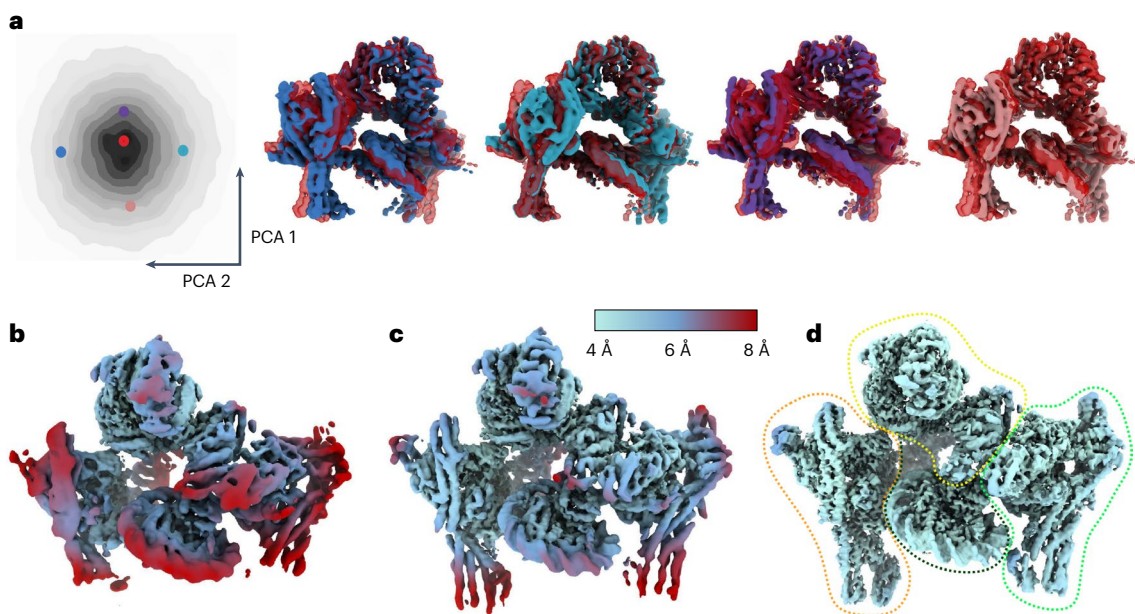

**Fig. 5 | DynaMight results for the complete kinetochore complex. a**, PCA of the conformational space (on the left) with highlighted positions of five conformation states, the maps of which are shown in the same colors on the right. (Only the latent space for one of the two half sets is shown.) **b**, Maps from standard RELION consensus refinement. **c**, DynaMight reconstruction. **d**, Reconstruction using RELION multi-body refinement. The outline regions in the latter show the four bodies that were used for multi-body refinement. The maps in **b**–**d** are colored by local resolution, as indicated by the color bar.

in the main text, and recently reviewed in ref. 28, multiple approaches from different laboratories have been proposed. In this paper we present an approach, called DynaMight, which consists of two VAEs that are trained independently on half sets to estimate displacements of a Gaussian model and a modified weighted backprojection algorithm to correct for the estimated deformations. To avoid deformations being described by the disappearance of Gaussians in one place and the appearance of Gaussians in another, and to limit the number of model parameters, DynaMight does not refine an occupancy factor for each Gaussian. Consequently, DynaMight cannot model compositional heterogeneity and it is unclear how it will perform on datasets with such heterogeneity. Compositional heterogeneity should thus be removed

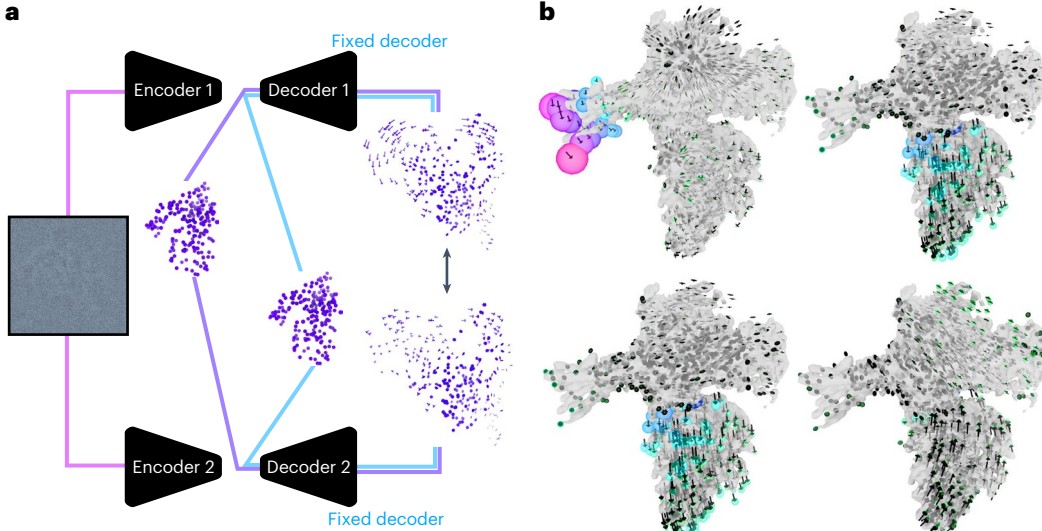

**Fig. 6 | Error estimation for the deformations. a**, Particles of a validation subset (here 10% of the particles) are fed into both encoders. The encoders are updated, whereas these images are not used for training the decoder. At evaluation time, both decoders can be evaluated for the consensus models (purple for the consensus model of half set 1 and blue for the consensus model of half set 2).

The resulting displacements can be compared. **b**, Example deformation fields for four particles. The radius of the sphere (colored by size from blue to pink) at the end of the deformations (black arrows) is determined by the norm of the difference of the deformations from the two decoders.

using existing discrete classification methods[1] before the application of DynaMight. We show for two datasets on the yeast inner kinetochore that DynaMight is useful in improving cryo-EM maps of macromolecular complexes that exhibit large amounts of flexibility, although scope remains for further improvements, of DynaMight in particular and how to deal with continuous structural heterogeneity in general.

Because of the high levels of experimental noise and the large number of parameters needed to describe continuous structural flexibility in the particles, an obvious way to improve these methods is the incorporation of prior knowledge. However, our results on the spliceosomal B complex show that such approaches are not without risk. We observe that there are enough parameters in DynaMight's neural networks to result in deformation fields that, when used in deformed backprojection, will reproduce incorrect features from the consensus model that is used to regularize these deformations. That model bias may play a role is perhaps not surprising, given that similar observations have been made for standard (structurally homogeneous) refinement, where only five parameters (three rotations and two translations) are used for every particle. The total number of parameters in DynaMight's VAE is approximately 10 million, which results in considerably higher numbers of parameter per particle for typical datasets. We do not believe that the risk of overfitting exists only in DynaMight. Other approaches that describe structural heterogeneity in the dataset with large neural networks, or other approaches with high numbers of parameters per particle, such as cryoDRGN[9], Zernike3D[13] and 3DFlex[12], will probably also be susceptible to these problems. The development of validation procedures will thus be important. In DynaMight, we chose not to expose the usage of atomic models for regularization of the deformations to the user, as potential model bias toward those models takes away the possibility to validate the map by the appearance of protein-like features. The exploration of more sophisticated methods, where part of the information of atomic models is used and other parts are set aside for validation, may yield better methods, while still allowing proper validation.

Because model bias may affect the estimation of deformation fields, over-estimation of the resolution of reconstructions that correct for these deformations may represent another pitfall. Resolutions are typically measured by Fourier shell correlation between two half sets. However, if deformations have been estimated jointly for both half sets, with the same reference map as origin, then incorrect features from

the reference model may be reproduced in both half-reconstructions, resulting in inflated Fourier shell correlation curves and over-estimation of resolution. Our results with the yeast kinetochore complex (Extended Data Fig. 5) indicate that 3DFlex[12] may suffer from such over-estimation of resolution. By training two independent VAEs with separate consensus models for both half sets, similar to 'gold-standard' approaches in standard refinement[29,30], this risk is avoided in DynaMight.

Training two VAEs independently on two half sets of the data also offers an opportunity to estimate the uncertainty in the estimated deformations. Although in recent years multiple methods have been proposed to analyze molecular motions in cryo-EM datasets, less consideration has been given to what extent these motions can be trusted. Error estimates on the deformations can be obtained for a subset of the particles (we used 10% in Fig. 6), by excluding this subset from the training of the decoders and only using it for training its embedding to latent space. For each particle in this subset, one obtains an embedding with both separate encoders to obtain a latent representation for the corresponding decoder. Applying both decoders to get the displacements of either of the consensus models then leads to two independent estimates of the deformations for the particles in the subset. The difference between these two estimates provide an estimate of the errors in them. We illustrate this procedure in Fig. 6b, where we observe that the errors in the deformations vary among particles and among different regions of the CCAN–CENP-A complex. Future developments in regularization methods as described above may benefit from considering estimated errors in the deformations.

Besides estimation of deformations, DynaMight also implements a reconstruction algorithm that aims to correct for the deformations through the reconstruction of an improved consensus map. Reconstruction via equation (7) only gives an approximation of the minimizer of the convex problem in equation (6). Although it is therefore not guaranteed to yield a useful solution, in practice we observe that DynaMight results in maps with improved local resolutions compared to the standard RELION reconstruction algorithm that assumes structural homogeneity. The improvements in the reconstructed maps provide some level of validation of the estimated deformation fields. Nevertheless, our observations that multi-body refinement yields better local resolutions for the complete inner kinetochore complex suggest that there is room for further improvement. It is possible that iterative real-space methods,

such as those implemented in 3DFlex[12] or Zernike3D[13], may yield better results. But the iterative approaches would be even more computationally expensive than our weighted backprojection approach, as they may require multiple sweeps through the data and optimization of hyperparameters, such as the step size. Alternatively, the results with multi-body refinement suggest that it may be possible to divide each particle into many smaller 'bodies', and to insert Fourier slices of each of these bodies using orientations that are a combination of the consensus orientation and the average deformation field at that region.

Although opportunities for further improvements exist, we believe that the current implementation of DynaMight will already be useful. Unlike multi-body refinement, there is no need for the design of masks that delineate the bodies. In fact, analysis of deformations estimated by DynaMight may assist users to define those masks for subsequent multi-body refinements. The implementation inside RELION-5 will make DynaMight easily accessible to many users, and its wider application will provide feedback for future developments of even better tools to analyze molecular motions in biological macromolecules. The unresolved challenges, as explored in this paper, of how to exploit more previous knowledge, while preventing the pitfalls of model bias, and how to validate the estimated deformations, imply that this topic will remain an active area of research.

## Online content

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

## Methods

### Initialization of the reference model

We model the 3D cryo-EM density map $f : \mathbb{R}^3 \to \mathbb{R}$ by a sum of $N_g$ Gaussian functions. The density $f$ is defined by

$$f(\mathbf{x}) = \sum_{j=1}^{N_c} \left( \sum_{i=1}^{N_g} d_{j,i} a_j \exp\left( \frac{\| \mathbf{x} - c_i \|}{s_j} \right) \right). \tag{8}$$

Here $N_c$ is a fixed number defining how many distinct widths are used in the Gaussian model. For the $i$th Gaussian the vector $\mathbf{d}_{-,i}$ satisfies $\sum_{j=1}^{N_c} d_{j,i} = 1$ and $d_{j,i} \geq 0$ for all $j \in \{1, \ldots, N_c\}$. This weight vector continuously classifies the type of Gaussian that is selected for a certain position of the Gaussian model. Although we used $N_c = 1$ in all our results, using more classes could be helpful for cases where the consensus map contains large variations in local resolution and the same width of all Gaussians does not give a reasonable representation of the map. The learnable parameters in this model are the widths $(s_1, \ldots, s_{N_c})$, composition vectors $\mathbf{d}$ and amplitudes $(a_1, \ldots, a_{N_c})$. These parameters are optimized globally, meaning that they are independent of the projection image, and stay the same over the whole dataset. Whereas a per-Gaussian amplitude parameter would be possible and would enable the representation of compositional heterogeneity, we decided to use the same amplitudes for all Gaussians. The reason for this is that otherwise movement could also be represented by Gaussian densities vanishing and reappearing at different places. We call the parameters $(\mathbf{a}, \mathbf{s}, \mathbf{d}, \mathbf{c})$ of the Gaussian model the reference parameters and we use a separate optimizer (ADAM) to update them. The total number of reference parameters is $N_g \times 3 + N_c \times (N_g + 2)$. For our experiments, we used only one class of Gaussians, resulting in $N_g \times 3 + 2$ parameters. The consensus model serves as the starting point for the decoder that predicts how every Gaussian in the model moves to explain the corresponding experimental image.

In the recommended way of running DynaMight, the initial reference map, that is, the reconstruction from the consensus refinement, is thresholded and randomly filled with $N_g$ Gaussians that are within the region of the map exceeding this threshold. The threshold should be chosen such that density in the flexible regions remains, but no noise is visible in the solvent region. The parameters $a$ and $s$ are initialized to reasonable numbers such that the norm of the Gaussian model equals the norm of the consensus reconstruction and the classification weights are initialized randomly. Once the reference parameters are initialized, we optimize the reference model using gradient descent (that is, without any networks), minimizing the mean squared error to the experimental images.

Alternatively, Gaussians may be initialized from the positions of an atomic model that is rigid-body fitted into the consensus map. For our experiments with atomic models for the spliceosome dataset, we used the deposited atomic model (PDB 5nrl). Instead of using one Gaussian per atom, we coarse-grained the atomic models. For every amino acid we used one main chain Gaussian that was located at the Bary center of the N, C and O atoms. Subsequent main chain Gaussians were connected by an edge in the graph used for regularization. The number of Gaussians used to represent the side chains varied for different amino acids. We placed one additional Gaussian at the Bary center of the $\alpha$, $\beta$ and $\gamma$ position side-chain atoms of all amino acids, except for 'PRO', where we took the Bary center of atoms at the $\alpha$, $\beta$, $\gamma$ and $\delta$ positions, and for 'SER', 'CYS', 'ALA', 'GLY', 'VAL' and 'THR', where we placed a Gaussian at the $\beta$ position. For larger amino acids, we placed additional side-chain Gaussians at the Bary center of the remaining side-chain atoms, except for 'TYR' and 'TRP', where we used two additional Gaussians. Subsequent Gaussians from the side chains were connected to each other and then to the corresponding main chain Gaussian with edges for the regularization functional. The amplitudes of the Gaussians were chosen to be proportional to the combined atomic number of all (nonhydrogen) atoms grouped together for the corresponding Gaussian. For nucleic

acids we used four Gaussians: one at the phosphate position and one at the Bary center of the sugar form the main chain of the nucleic acid chain and two Gaussians at the bases. Again, the amplitudes were set to be proportional to the combined atomic number within each group.

### The VAE

A VAE estimates displacements of the Gaussians from the reference model. An encoder learns an embedding to a low dimensional latent space that describes the conformational landscape of the dataset. The decoder estimates a deformation, given a point in that latent space and a position in the 3D reference.

The input to the encoder is a flattened (real-space) experimental image $y_i$ and the output are two vectors $(\mu_i, \sigma_i) \in \mathbb{R}^{N_l} \times \mathbb{R}^{N_l}$, which describe the mean and standard deviation used to generate a sample, which serves as an input for the decoder. The encoder is a fully connected neural network with three linear layers and rectified linear unit activation functions. To optimize the weights of the encoder we used the ADAM optimizer with a learning rate of 0.001. We tried to use alternative encoder architectures using residual connections, more linear layers and convolutional neural networks, but without observing relevant improvements in performance. Even when substituting the input images with a different unique signal (we used a random vector per image), the deformations are not worse. We conclude that the encoder does not effectively use the information that is present in the images, suggesting that one could optimize the latent representation itself via an autodecoder[12].

The decoder is at the heart of our approach. Given a conformational representation it estimates a deformation for the corresponding particle image. It takes the latent representation $z_i$ and a spatial position, and outputs the displacement of that which is predicted at this spatial position. During training, the positions where the decoder is evaluated are the Gaussian positions in the reference model. Compared to ref. 10 we use a coordinate-based network that takes the input position as an input. To augment the 3D coordinates, we use positional encoding with ten encoding dimensions, which has shown to resolve higher resolution information in coordinate-based networks[31]. We use the sine and cosine function for lifting the 3D position to a higher dimensional space as described in ref. 32. We observed that without the positional encoding of the input coordinate the deformations are too smooth and that localized motion is not captured well. The use of a coordinate-based network results in a network that approximates a deformation field that can be evaluated at any position in $\mathbb{R}^3$.

The decoder itself is a fully connected network $\delta$ with exponential linear unit (ELU) activation functions and an additive residual connection (Extended Data Fig. 1). We use eight linear layers to obtain for a given spatial position $\mathbf{x} \in \mathbb{R}^3$ the deformed position:

$$\mathcal{D}(z_i, \mathbf{x}) = \mathbf{x} + \delta(z_i, \mathbf{x}). \tag{9}$$

In the training phase, we evaluate the decoder for all the positions $\mathbf{c}_0$ in the reference model. We then model the forward operator of cryo-EM by projecting the center points of the deformed Gaussian reference model using the orientation of the particle, resulting in 2D coordinates $\xi_i$. These coordinates are then placed into an (oversampled) 2D grid using bilinear interpolation. Then we compute the 2D Fourier transform, approximating the Fourier transform of the sum of deltas. Subsequently, we multiply the resulting Fourier-space image with the Gaussian basis function $G_s$ and the CTF $\mathcal{C}_i$ resulting in the projection image $g_i$ of the deformed Gaussian model

$$g_i \approx \mathcal{F}\left( \sum_{j=1}^{N_g} a \delta_{\xi_i^j} \right) \cdot \mathcal{C}_i \cdot G_s. \tag{10}$$

If more than one type of Gaussian exists, the same operation is repeated for all types and weighted by the class assignment vector $\mathbf{d}$.

The resulting reference projection image $g_i$ is then compared to the experimental image, using a mean squared error as the loss function (also below).

## Training

After initialization of the Gaussians in the consensus reconstruction, during the first epochs (that is, sweeps over the two half sets for both models) of training we only optimize the Gaussian parameters, that is their widths, amplitudes and positions. After this initial phase, we also start optimizing the network parameters of the two independent VAEs, which are initially assigned random values. Both phases of training use the ADAM optimizer at a learning rate 0.0001.

To get physically meaningful deformations, the reference model itself should lie within the distribution of all the conformations estimated by the decoder, rather than being a nonexisting average of conformations (as the reconstruction from the consensus refinement is). To achieve this, we apply two heuristic strategies that gradually improve the reference model. First, after every 30 epochs, we fix the encoder and decoder for five epochs and only adjust the Gaussian parameters. Second, at every tenth epoch where the decoder is not fixed, we replace the positions of the Gaussians of the reference model by the predicted Gaussian positions with the smallest displacement from the current reference model. The latter ensures that the reference model is in the distribution of deformed models. Without this replacement strategy, we observed that the reference model can move out of distribution, sometimes even to a point where the structure is completely distorted. As long as the deformations satisfy the regularization constraints, this should not change the value of the loss function, but we observed that this can lead to unphysical displacements of the Gaussians and suboptimal reconstructions. To also ensure that the reference models of the two independent half sets are in the same conformation, we generate a binary mask around the Gaussians positions of one half set and substitute the Gaussian positions of the other half set with the average over 100 predictions where the number of Gaussians inside this mask is the highest. The binary mask covers all voxels that have a Gaussian within a distance of 6 Å from the voxel center. Fourier shell correlations of the Gaussian model to the consensus and the final Gaussian model to the final reconstruction are displayed in Supplementary Fig. 1.

Training is stopped when the updates of the consensus model do not yield improvements in the data loss mean squared error (MSE; below) anymore. More specifically, we stop training if the MSE loss increased for the $k$th time. In our experiments we used the default value of $k = 40$.

**Loss functions and regularization.** Denoting by $g_i$ the reference projection image generated by the current VAE, the main loss function is the data loss, which for a batch $\mathcal{B} := (g_i, y_i)_{i \in \mathbf{B}}$ is computed in Fourier space as

$$\mathcal{F}(\mathcal{B}) := \frac{1}{|\mathcal{B}|} \sum_{i \in \mathcal{B}} \| g_i - y_i \|_{\Sigma}^2, \tag{11}$$

where the resolution-dependent noise weights $\Sigma$ are estimated by the radially averaged power of the error on a subset of particle images.

Auxiliary losses are used to regularize the deformations of the Gaussian model. In the recommended way of running DynaMight, a graph is constructed by connecting Gaussians that are within a certain distance with edges. The set of edges is defined by

$$E_{ij} = \begin{cases} 1 & \| c_i - c_j \| < 1.5 \, c_{\text{mean}}, \\ 0 & \text{else.} \end{cases} \tag{12}$$

Here $c_{\text{mean}}$ is the mean distance in the graph $F_{ij}$, which is created by connecting every point to its two nearest neighbors. These graphs are recalculated from the reference model after every epoch. For the

deformation of the $k$th image $\Gamma_k$ the following regularization functional then preserves distances after displacement, enforcing local isometry:

$$\mathcal{R}_d(\Gamma_k) = \sum_{\{(i,j):E_{ij}=1\}} \left| \| c_i - c_j \| - \| \Gamma_k(c_i) - \Gamma_k(c_j) \| \right|^2, \tag{13}$$

Additionally, we use a repulsion loss penalizing Gaussians that are too close to each other

$$\mathcal{R}_r(\Gamma_k) = \sum_{\{(i,j):E_{ij}=1\}} \chi_{\| \Gamma_k(c_i) - \Gamma_k(c_j) \| < \tau}(\| \Gamma_k(c_i) - \Gamma_k(c_j) \| - \tau)^2, \tag{14}$$

where $\chi_{\| \Gamma(c_i) - \Gamma(c_j) \| < \tau} = 1$ if the distance between neighboring Gaussians is less than $\tau$. We set $\tau$ to $c_{\text{mean}}$ for all our results.

The total loss function is then given by

$$\mathcal{L}(\mathcal{B}) = \mathcal{F}(\mathcal{B}) + \lambda \frac{1}{|\mathcal{B}|} \sum_{i \in \mathcal{B}} [\mathcal{R}_d(\Gamma_i) + \mathcal{R}_r(\Gamma_k)] =: \mathcal{F}(\mathcal{B}) + \lambda \mathcal{R}(\mathcal{B}).$$

The parameter $\lambda$ is a dynamic regularization parameter that is recalculated after every epoch. We do that by calculating the norm of the gradients of both loss terms $\mathcal{L}$ and $\mathcal{R}$ and define $\lambda$ such that the ratio of these norms equal a user-defined number. When set to 1 the norm of the gradient of both terms is equal. For all our results we set this value to 0.9, which results in slightly more influence of the data term $\mathcal{L}$.

For the results, where we used the coarse-grained atomic model as a reference, we used the same data loss function $\mathcal{F}$ (equation (12)), but in contrast to the above described heuristic method to construct the edges between the Gaussian, the graph $E$ is obtained from the coarse graining of the atomic model. The regularization that preserves distances is applied in the same way (equation (13)) with the fixed graph from the coarse graining. The second regularization functional (equation (14)) is not used in this case, since the distances in the reference model are fixed.

## Improved reconstruction

To calculate an improved reconstruction from the estimated deformations, we use a network $\mathcal{D}^{-1}$ with the same architecture as the decoder to estimate a deformation field that maps back a deformed position to its original location. Again this network is coordinate-based and can be evaluated on an arbitrary position $\mathbf{x} \in \mathbb{R}^3$. Given the latent representation of each particle we train the neural network $\mathcal{D}^{-1}$ to map back the positions predicted by the trained VAE to the positions of the reference model. Since the model should estimate the inverse deformation of the decoder $\mathcal{D}$, it should satisfy

$$\mathcal{D}^{-1}\left( \mu_i, \mathcal{D}\left(z_i, c_j^0\right)\right) = c_j^0.$$

For each image $g_i$ the neural network takes as input the latent representation $\mu_i$ from the previously trained encoder $\mathcal{E}$ and a positional encoding of the deformed Gaussian positions $\mathcal{D}(z_i, \mathbf{c}^0)$. The concatenated positional encoding and latent representation are then mapped by an multilayer perceptron with six layers and a single additive residual connection to the original coordinates of the consensus model $\mathbf{c}^0$. The loss function is the $L^2$ distance between the positions

$$\frac{1}{N_d N_g} \sum_{i=1}^{N_d} \sum_{j=1}^{N_g} \left\| \mathcal{D}^{-1}\left( \mu_i, \mathcal{D}\left(z_i, c_j^0\right)\right) - c_j^0 \right\|^2$$

We optimized the weights of the inverse deformation network for 200 epochs with the ADAM optimizer for all our results. Once the network has been trained, the backprojection algorithm evaluates it for the latent representation of every particle on a 3D grid and applies the deformation to the CTF-multiplied, backprojected image. For computational speed, we evaluated the inverse deformation on a two

times coarser grid, and then up-sampled the deformation fields to the original box size again using bilinear interpolation. The resulting volumes are then summed up and divided by the backprojected squared CTFs as illustrated in Fig. 1.

## Reporting summary

Further information on research design is available in the Nature Portfolio Reporting Summary linked to this article.

## Data availability

The authors declare that there are no restrictions on data availability. The datasets of the CCAN–CENP-A complex and of the yeast inner kinetochore complex bound to a CENP-A have been deposited at EMPIAR[19] and are available under accession codes EMPIAR-11890 and EMPIAR-10073, respectively. The local-resolution filtered reconstructions and the DynaMight half-maps for all datasets are available on Electron Microscopy Data Bank under the accession codes EMD-19791 for the precatalytic spliceosome, EMD-19789 for the tri-snRNP complex, EMD-19799 for the CCAN–CENP-A complex and EMD-19794 for the yeast inner kinetochore complex bound to CENP-A.

## Code availability

DynaMight is distributed for free under a Berkeley Software Distribution (BSD) license and can be downloaded from https://github.com/3dem/DynaMight. It is installed automatically with RELION-5.

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

## Acknowledgements

We thank J. Grimmett, T. Darling and I. Clayson for help with high-performance computing. C. Esteve Yagüe, W. Dieperveen, C. Schönlieb, O. Öktem and K. Jamali for helpful discussions, and D. Barford for critical reading of the paper. This work was supported by the Medical Research Council (MRC), as part of United Kingdom Research and Innovation (UKRI) (grant no. MC_UP_AO25_1013 to S.H.W.S.), and by Wave 1 of The UKRI Strategic Priorities Fund under the Engineering & Physical Research Council (EPSRC) grant no. EP/W006022/1, particularly the 'AI for Science' theme within that grant and The Alan Turing Institute. The contribution by T.D. was funded by Cancer Research UK (grant no. C576/A14109) and UKRI (grant no. MC_UP_1201/6) to D.K. For the purpose of open access, the MRC Laboratory of Molecular Biology has applied a CC BY public copyright licence to any author accepted manuscript version arising.

## Author contributions

J.S. designed and implemented DynaMight, ran all experiments and analyzed results. D.K. helped with the design and implementation of DynaMight. A.B. provided help with Python and Napari. T.D. and D.K. provided and analyzed yeast kinetochore data. S.H.W.S. provided help with RELION and supervised the project. All authors contributed to writing of the paper.

## Competing interests

The authors declare no competing interests.

## Additional information

**Extended data** is available for this paper at https://doi.org/10.1038/s41592-024-02377-5.

**Correspondence and requests for materials** should be addressed to Johannes Schwab or Sjors H. W. Scheres.

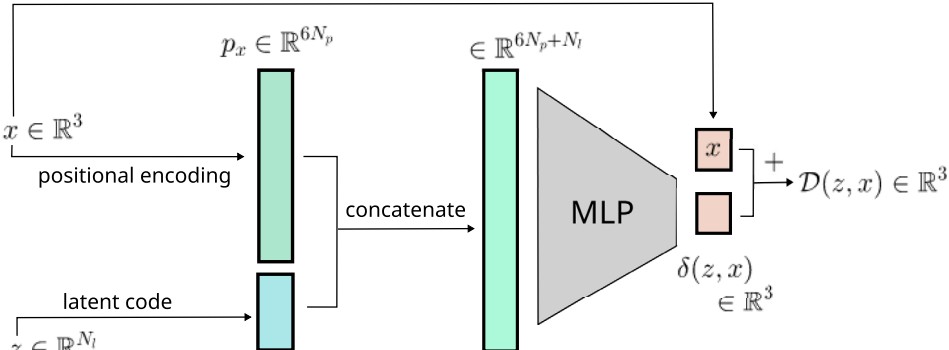

**Extended Data Fig. 1 | Diagram of the decoder architecture.** The queried position is lifted to a higher dimensional space via a fixed positional encoding function, where the lifting dimension is defined by Np. The Nl dimensional latent code is concatenated with the encoded position, and input to a multilayer perceptron (MLP), which outputs a 3-dimensional displacement vector. To obtain the final position the displacement vector is added to the original position x.

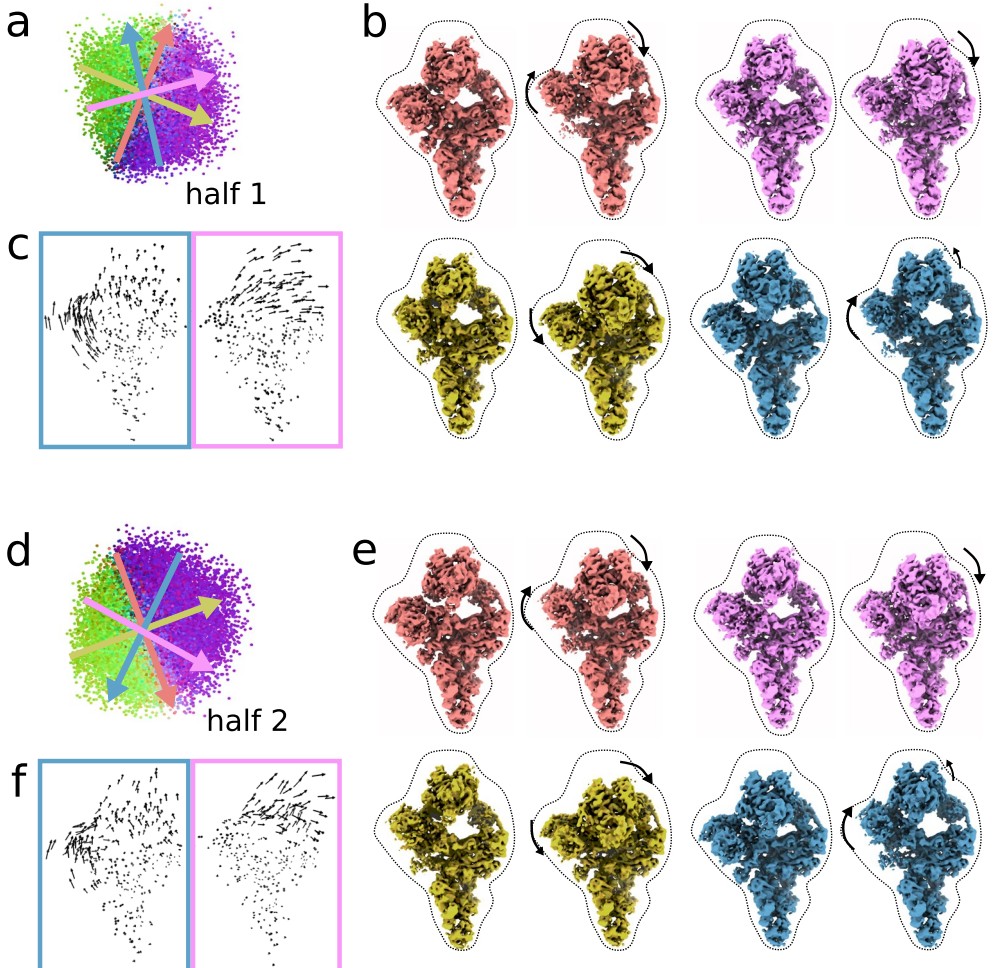

**Extended Data Fig. 2 | nalysis of motions for the pre-catalytic spliceosome. a,d)** Latent spaces of both half sets (half 1 and half 2) for the pre-catalytic spliceosome dataset EMPIAR-(10180) are coloured by the mean movement direction. Four different deformations are visualized as coloured (red, pink, yellow and blue) arrows from one point in latent space to another. **b,e)** The corresponding maps are shown in the same colours, with black arrows indicating the main deformations. **c,f)** For the blue and the pink deformations, 3D deformation fields are also shown as black arrows for the displacements of individual Gaussians. The latent spaces of the two half sets are organized in a similar way, with similar deformations along the shown directions. The observed motions are comparable to the ones obtained by e2gmm.

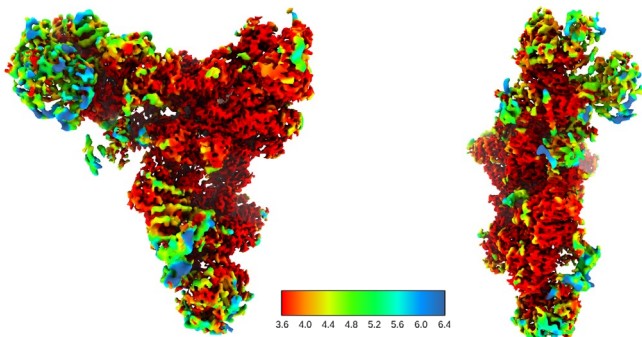

**Extended Data Fig. 3 | DynaMight reconstruction for the spliceosomal tri-snRNP complex.** The DynaMight reconstruction from 86,624 selected particles of data set EMPIAR-10073 is coloured by local resolution, as estimated using cryoSPARC. The map is displayed in two orthogonal orientations and with a local resolution colour scheme that matches the figure used to illustrate the 3DFlex method (ref. 12).

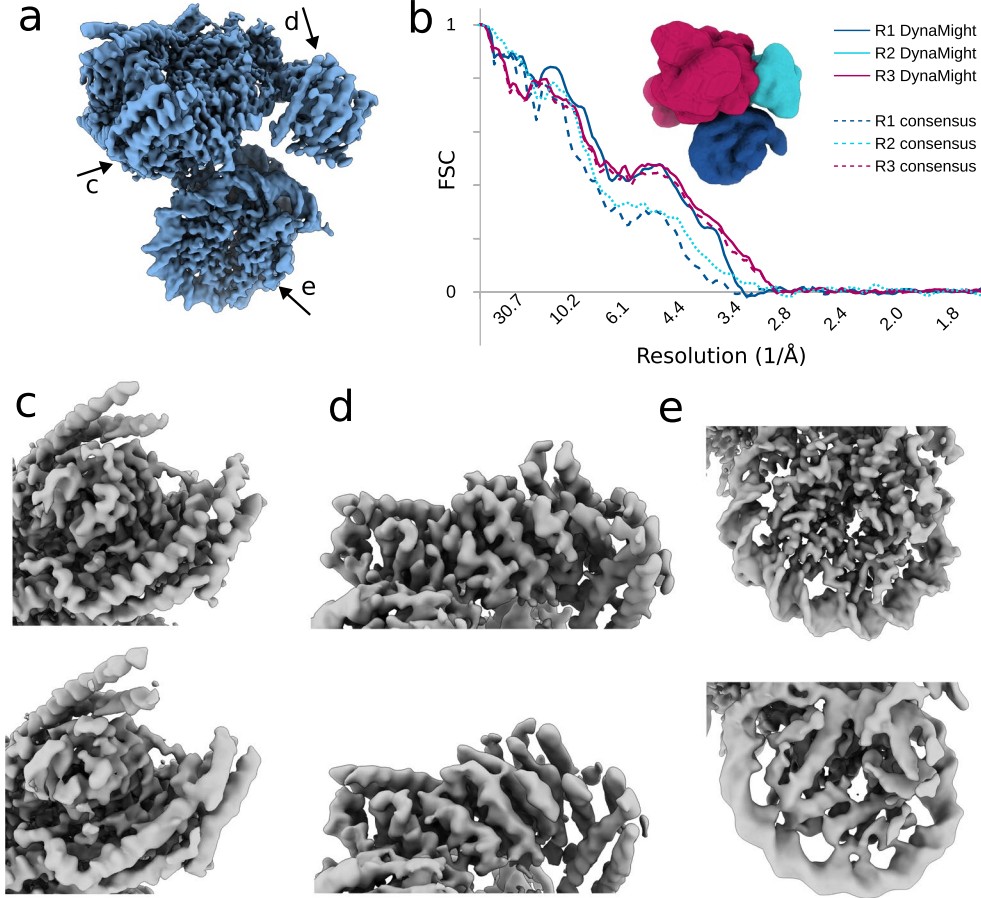

**Extended Data Fig. 4 | DynaMight reconstruction for the CCAN:CENP-A complex. a**) Local resolution filtered map of the DynaMight reconstruction. **b**) Fourier shell correlation (FSC) between atomic models fitted into 3 regions of the maps (R1-R3) and the DynaMight and consensus reconstruction.
**c-e**) Comparison of DynaMight (top) and consensus map (bottom) in the regions indicated with black arrows in panel a.

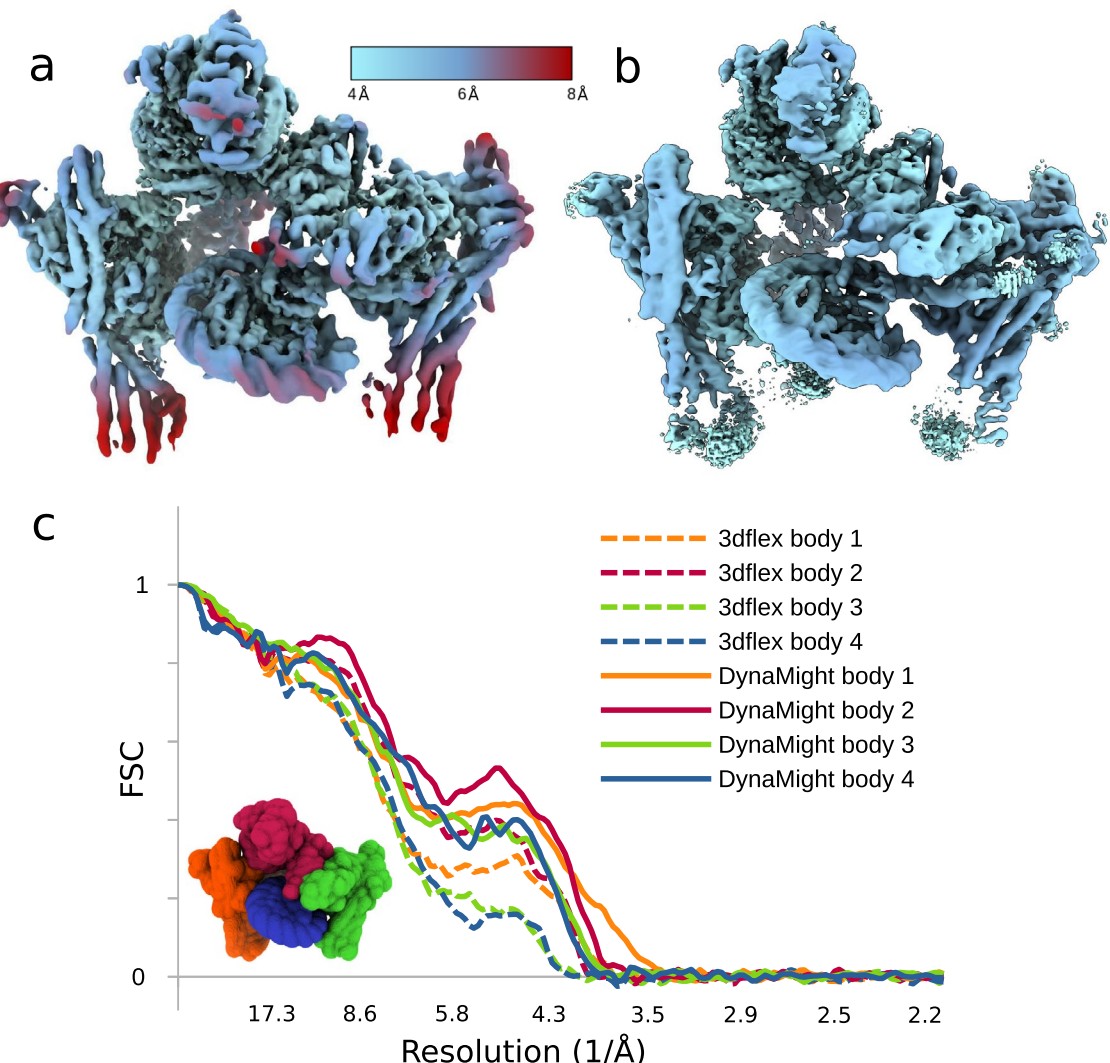

**Extended Data Fig. 5 | Comparison between DynaMight and 3DFlex for the kinetochore complex. a**,**b**) Reconstructions from DynaMight (**a**) and 3DFlex (**b**), coloured according to local resolution (as estimated in RELION) with the same colour scheme, ranging from cyan (4 Å) to red (8 Å). A 10-dimensional latent space was used for both methods; all other parameters were kept at default. **c**) Fourier shell corre-lation (FSC) between rigid-body fitted atomic models and the reconstructed maps for DynaMight (solid lines) and 3DFlex (dashed lines) for four domains of the kinetochore complex (body 1-4, in orange, red, green and blue, respectively).

# Reporting Summary

## Statistics

For all statistical analyses, confirm that the following items are present in the figure legend, table legend, main text, or Methods section.

| n/a | Confirmed | |
|---|---|---|
| ☐ | ☒ | The exact sample size (*n*) for each experimental group/condition, given as a discrete number and unit of measurement |
| ☒ | ☐ | A statement on whether measurements were taken from distinct samples or whether the same sample was measured repeatedly |
| ☒ | ☐ | The statistical test(s) used AND whether they are one- or two-sided *Only common tests should be described solely by name; describe more complex techniques in the Methods section.* |
| ☒ | ☐ | A description of all covariates tested |
| ☒ | ☐ | A description of any assumptions or corrections, such as tests of normality and adjustment for multiple comparisons |
| ☐ | ☒ | A full description of the statistical parameters including central tendency (e.g. means) or other basic estimates (e.g. regression coefficient) AND variation (e.g. standard deviation) or associated estimates of uncertainty (e.g. confidence intervals) |
| ☒ | ☐ | For null hypothesis testing, the test statistic (e.g. *F*, *t*, *r*) with confidence intervals, effect sizes, degrees of freedom and *P* value noted *Give P values as exact values whenever suitable.* |
| ☒ | ☐ | For Bayesian analysis, information on the choice of priors and Markov chain Monte Carlo settings |
| ☒ | ☐ | For hierarchical and complex designs, identification of the appropriate level for tests and full reporting of outcomes |
| ☒ | ☐ | Estimates of effect sizes (e.g. Cohen's *d*, Pearson's *r*), indicating how they were calculated |

*Our web collection on statistics for biologists contains articles on many of the points above.*

## Software and code

Policy information about availability of computer code

| Data collection | Data collection software is not relevant to this study |
|---|---|
| Data analysis | DynaMight is distributed for free under a BSD license and can be downloaded from https://github.com/3dem/DynaMight. It is installed automatically with RELION-5.0 |

For manuscripts utilizing custom algorithms or software that are central to the research but not yet described in published literature, software must be made available to editors and reviewers. We strongly encourage code deposition in a community repository (e.g. GitHub). See the Nature Portfolio guidelines for submitting code & software for further information.

## Data

Policy information about availability of data

All manuscripts must include a data availability statement. This statement should provide the following information, where applicable:
- Accession codes, unique identifiers, or web links for publicly available datasets
- A description of any restrictions on data availability
- For clinical datasets or third party data, please ensure that the statement adheres to our policy

The authors declare that there are no restrictions on data availability. The data sets of the CCAN:CENP-A complex and of the yeast inner kinetochore complex bound to a CENP-A have been deposited at EMPIAR and are available under accession codes 11890 and 10073, respectively. Data sets of the spliceosome complexes were already available under accession codes 10180 and 10073. Atomic coordinates from PDB entries 5nrl, 1g88 and 7yuy were also used in this study. The local

resolution filtered reconstructions and the DynaMight half-maps for all datasets are available on EMDB under the accession codes EMD-19791 for the pre-catalytic spliceosome, EMD-19789 for the tri-snRNP complex, EMD-19799 for the CCAN:CENP-A complex and EMD-19794 for the yeast inner kinetochore complex bound to CENP-A.

## Human research participants

Policy information about studies involving human research participants and Sex and Gender in Research.

| | |
|---|---|
| Reporting on sex and gender | N/A |
| Population characteristics | N/A |
| Recruitment | N/A |
| Ethics oversight | N/A |

Note that full information on the approval of the study protocol must also be provided in the manuscript.

# Field-specific reporting

Please select the one below that is the best fit for your research. If you are not sure, read the appropriate sections before making your selection.

☒ Life sciences          ☐ Behavioural & social sciences          ☐ Ecological, evolutionary & environmental sciences

For a reference copy of the document with all sections, see nature.com/documents/nr-reporting-summary-flat.pdf

# Life sciences study design

All studies must disclose on these points even when the disclosure is negative.

| | |
|---|---|
| Sample size | Cryo-EM data set sizes were determined by the publicly available data sets or the experimentalists providing the data. They were originally probably determined by the amount of available microscopy time. The number of data sets described in this study was deemed sufficient to illustrate the general usefulness of the DynaMight approach. |
| Data exclusions | We selected a subset of 45k spliceosome particles from the EMPIAR-10180 data set in order to make it compositionally homogeneous. We used standard 3D classification in RELION to make this selection, as DynaMight cannot handle structural heterogeneous data sets. |
| Replication | No replication experiments were performed, as noise on the outcome of the computational analysis was not considered to be affecting the conclusions. |
| Randomization | For each data set, two random half-sets were employed for resolution estimation. Randomisation was performed using a random number generator. |
| Blinding | No blinding was performed as bias from the user is not expected to affect the outcome of the computational analysis. |

# Reporting for specific materials, systems and methods

We require information from authors about some types of materials, experimental systems and methods used in many studies. Here, indicate whether each material, system or method listed is relevant to your study. If you are not sure if a list item applies to your research, read the appropriate section before selecting a response.

### Materials & experimental systems

| n/a | Involved in the study |
|---|---|
| ☒ ☐ | Antibodies |
| ☒ ☐ | Eukaryotic cell lines |
| ☒ ☐ | Palaeontology and archaeology |
| ☒ ☐ | Animals and other organisms |
| ☒ ☐ | Clinical data |
| ☒ ☐ | Dual use research of concern |

### Methods

| n/a | Involved in the study |
|---|---|
| ☒ ☐ | ChIP-seq |
| ☒ ☐ | Flow cytometry |
| ☒ ☐ | MRI-based neuroimaging |

