## [Peer Review File · Nature Methods]

Peer Review Information

Manuscript Title: DynaMight: estimating molecular motions with improved reconstruction from cryo-EM images

Corresponding author name(s): Sjors Scheres

Editorial Notes: None

Reviewer Comments & Decisions:

Decision Letter, initial version:

Dear Sjors,

Your Article, "DynaMight: estimating molecular motions with improved reconstruction from cryo-EM images", has now been seen by 2 reviewers. As you will see from their comments below, although the reviewers find your work of considerable potential interest, they have raised a number of fairly serious concerns. We are interested in the possibility of publishing your paper in Nature Methods, but would like to consider your response to these concerns before we reach a final decision on publication.

We therefore invite you to revise your manuscript to address these concerns. In particular we recommend that you compare DynaMight to other tools in this space and include a demonstration on a synthetic dataset as recommended by the reviewers. It would be important to show that your method has a strong advance, as well to provide/clarify all method details.

* include a point-by-point response to the reviewers and to any editorial suggestions

* please underline/highlight any additions to the text or areas with other significant changes to facilitate review of the revised manuscript

* address the points listed described below to conform to our open science requirements

* ensure it complies with our general format requirements as set out in our guide to authors at www.nature.com/naturemethods

* resubmit all the necessary files electronically by using the link below to access your home page

[REDACTED]

We hope to receive your revised paper within 10 weeks. If you cannot send it within this time, please let us know. In this event, we will still be happy to reconsider your paper at a later date so long as nothing similar has been accepted for publication at Nature Methods or published elsewhere.

OPEN SCIENCE REQUIREMENTS

REPORTING SUMMARY AND EDITORIAL POLICY CHECKLISTS

Please note that these forms are dynamic ‘smart pdfs’ and must therefore be downloaded and completed in Adobe Reader. We will then flatten them for ease of use by the reviewers. If you would like to reference the guidance text as you complete the template, please access these flattened versions at <http://www.nature.com/authors/policies/availability.html>.

DATA AVAILABILITY

All novel DNA and RNA sequencing data, protein sequences, genetic polymorphisms, linked genotype and phenotype data, gene expression data, macromolecular structures, and proteomics data must be deposited in a publicly accessible database, and accession codes and associated hyperlinks must be provided in the “Data Availability” section.

Please include a “Data availability” subsection in the Online Methods. This section should inform readers about the availability of the data used to support the conclusions of your study, including accession codes to public repositories, references to source data that may be published alongside the paper, unique identifiers such as URLs to data repository entries, or data set DOIs, and any other statement

about data availability. At a minimum, you should include the following statement: “The data that support the findings of this study are available from the corresponding author upon request”, describing which data is available upon request and mentioning any restrictions on availability. If DOIs are provided, please include these in the Reference list (authors, title, publisher (repository name), identifier, year). For more guidance on how to write this section please see: <http://www.nature.com/authors/policies/data/data-availability-statements-data-citations.pdf>

CODE AVAILABILITY

Please include a “Code Availability” subsection in the Online Methods which details how your custom code is made available. Only in rare cases (where code is not central to the main conclusions of the paper) is the statement “available upon request” allowed (and reasons should be specified).

MATERIALS AVAILABILITY

ORCID

Nature Methods is committed to improving transparency in authorship. As part of our efforts in this direction, we are now requesting that all authors identified as ‘corresponding author’ on published papers create and link their Open Researcher and Contributor Identifier (ORCID) with their account on

the Manuscript Tracking System (MTS), prior to acceptance. This applies to primary research papers only. ORCID helps the scientific community achieve unambiguous attribution of all scholarly contributions. You can create and link your ORCID from the home page of the MTS by clicking on 'Modify my Springer Nature account'. For more information please visit please visit www.springernature.com/orcid.

Sincerely,
Arunima

Arunima Singh, Ph.D.
Senior Editor
Nature Methods

Reviewers' Comments:

Reviewer #1:

Remarks to the Author:

Schwab et al. introduce DynaMight, a novel software tool designed for estimating a continuous range of conformations in cryo-EM datasets. This tool uses a method of learning 3D deformations from a Gaussian pseudo-atomic model for each particle image. These deformations are then reversed to enhance the reconstruction of the consensus structure. The authors demonstrate DynaMight's effectiveness on several experimental cryo-EM datasets and explain how to estimate errors in these deformations. Additionally, they distribute this tool as open-source software within RELION-5. The manuscript is well-written and meets high scientific standards. I recommend its publication, but I would like to raise a few points for consideration:

1. As the number of methods for addressing structural heterogeneity in cryo-EM is growing, comparing their results becomes essential. Specifically, comparing DynaMight with methods like e2gmm and 3dflex is vital to understand its position in this evolving field.

2. Analyzing synthetic data could be highly beneficial. Other approaches have used synthetic data to demonstrate their potential best-case outcomes and limitations. Since real-world data lacks a concrete ground truth, synthetic data can provide significant insights.

3. The manuscript quickly addresses compositional heterogeneity, stating that DynaMight cannot handle it and that it must be managed separately beforehand. However, removing this heterogeneity in biological samples is often challenging. I suggest a discussion on the consequences of running DynaMight on samples with compositional heterogeneity.

4. The decoder's architecture is not as clearly described as the encoder's. Details about the training process, such as whether the system uses pre-trained weights or initializes them for each run, would be helpful. A comprehensive methods section detailing the implementation would be beneficial.

5. The authors caution against using model regularization, but include it as an option in the software as far as I understand the text. However, I don't see it in the GUI as an option. Is it retained? If it is: Since there's no absolute ground truth in this field, and it's difficult to assess the extent of model bias, the rationale for retaining this potentially problematic option should be clarified. Perhaps comparing its use with less biased techniques, like multi-body refinement, could offer more clarity.

If it isn't: just clarify the text.

Minor comment:

6. The visualization of the model fit in the density shown in Figure 3 could be improved. Its current form is not very informative, and alternative ways of representation might be more effective.

Reviewer #2:

Remarks to the Author:

In this manuscript, the authors presented a novel protocol that combines the Gaussian representation of protein heterogeneity analysis (similar to Chen 2021), and a deformation field that utilizes the learned heterogeneity information to improve the quality of single particle reconstruction (similar to Herreros 2023). A VAE is used to learn the heterogeneity from the particles, and a separate neural network is used to bridge the Gaussian and the Cartesian grid representation and generate the deformation field. While the idea is conceptually attractive, the manuscript seems somewhat unfinished, with many details of the method implementation and the examples left out. This makes it difficult to judge the design of the method and the performance of the protocol.

My major comments are listed below.

Many details on the method implementation and the processing of the experimental datasets are missing. For example, what is the loss function used during training? Is the resolution of the reconstruction used to regularize the training and prevent overfitting? How are the weights of regularization determined? The encoder is described to be a 3-layer fully connected network, but the structure of the decoder is not presented. In the examples shown in the paper, it is unclear how many Gaussian functions were used to represent the molecule, how well does the Gaussian representation fit the initial reconstruction, or what is the scale of the movement (how many Å a domain moves or degree it tilts). It would be good to have a Method section and supplementary materials to describe the details, otherwise it is hard to tell what are the differences from the existing methods and what are the improvements.

The authors show potential model bias when information from atomic models were used for the analysis, but it is unclear what sort of constraints were applied. In Figure 2d, it seems that the atomic model was used to build the Gaussian model, and the distances between the bonded atoms were used to regularize the movement. How exactly is the Gaussian model initialized? Is the atomic model simply rigid-body fitted into the consensus map, or some sort of flexible fitting or Gaussian model optimization was performed? Is there any precaution taken for the gold-standard particle splitting, or the same atomic model was used for the analysis of the two half sets? How are the bond constraints applied? Is it determined from one existing model, or the ideal bonds length and angle from PDB statistics? How is the weighting between map and model regularization determined? For Figure 2b-c, it is mentioned that the bond constraints were not used, but how were the Gaussian model initialized in either case? If they were initialized the same way as Figure 2d, then obviously the same model bias gets into the system. If seeded directly from the volume, then it is hard to argue whether the difference comes from the Gaussian placement or the bond constraints. Overall, there is too little detail in the manuscript to decipher what is actually happening.

In the last example, the authors showed that the deformation based reconstruction is not performing as well as the multi-body refinement. Then in discussion, the authors argued there is a minimum size requirement for multi-body refinement, but not in the deformation based approach, so that the proposed method is still useful. However, none of those is shown experimentally. It is unclear what is the minimum size requirement for multi-body refinement, and there is no evidence that dynamight does not have that disadvantage. To truly show the new method is useful, it would be critical to have examples that multi-body does not work but the new method does, and discuss what is causing the differences.

Some less major comments:

For the encoder, it seems that the authors decided to use a fully connected network for real space images. This is quite unexpected, since convolutional networks for images have become the mainstream for many years now. Is there any specific concern against convolutional networks? Usage of fully connected networks might cause overfitting, since it is hard to imagine it capturing translation, rotation and conformation of each particle from the image. One way to check how much overfitting the encoder-decoder introduces is to input translated/rotated copies of the same particle and see if it gets mapped to the same position in the latent space. Including a separate test dataset during training may help too.

The authors mentioned the half-set splitting approach for heterogeneity validation, but in all three examples, only one latent space and conformations were shown (figure 5a for example). Are the conformations from one of the half sets? Are there any differences in the movement modes from the other half set? Is there a way to “align” the latent space from one VAE on one half set to another VAE’s output?

There is no clear indication of where the datasets used in the second and third examples come from. Are they open datasets from EMPIAR, or private datasets that may only be available upon request? In data availability, it only mentioned there is no restriction but not where to find the data.

Reviewer #3:

None

Author Rebuttal to Initial comments

We thank both reviewers for their constructive comments. Below, we answer each of their points in blue. We also note that since our initial submission we have made further improvements to DynaMight. Relatively minor changes in how we update the reference model during training (which are outlined in the Training subsection of the Extended Methods online section) have led to considerable improvements in the densities shown in Figures 2C and 4C of our revised manuscript.

Reviewer 1:

Schwab et al. introduce DynaMight, a novel software tool designed for estimating a continuous range of conformations in cryo-EM datasets. This tool uses a method of learning 3D deformations from a Gaussian pseudo-atomic model for each particle image. These deformations are then reversed to enhance the reconstruction of the consensus structure. The authors demonstrate DynaMight’s effectiveness on several experimental cryo-EM datasets and explain how to estimate errors in these deformations. Additionally, they distribute this tool as open-source software within RELION-5.

The manuscript is well-written and meets high scientific standards. I recommend its publication, but I would like to raise a few points for consideration:

1. As the number of methods for addressing structural heterogeneity in cryo-EM is growing, comparing their results becomes essential. Specifically, comparing DynaMight with methods like e2gmm and 3DFlex is vital to understand its position in this evolving field.

In response to this comment, we first compared DynaMight to e2gmm. The authors of e2gmm used the same EMPIAR-10180 data set as we use in our work. However, unlike DynaMight, e2gmm does not use the estimated deformations to calculate an improved reconstruction. The comparison between DynaMight and e2gmm is thus limited to a qualitative comparison to the motions that are described in the e2gmm paper. We have added a new Extended Data Figure 1 to the revised paper to illustrate that DynaMight yields similar motions to e2gmm. Our figure has been designed to look similar to Figure 4 of the e2gmm paper, which for ease of comparison we have copied in Figure 1 below. The 3DFlex method also uses the learnt deformations to calculate an improved reconstruction. We compared such improved reconstructions with those from DynaMight in two different ways. First, we applied DynaMight to the same dataset on the spliceosomal tri-snRNP complex that was used in the 3DFlex publication (EMPIAR-10073). The new Extended Data Figure 2 shows the DynaMight reconstruction in the same orientations as those shown in the 3DFlex paper (and with local resolution estimates also calculated by cryoSPARC). To aid the reviewer in this comparison, in Figure 2 below, we have also copied the corresponding figure from the 3DFlex publication.

Figure 1: Heterogeneity analysis of EMPIAR-10180 by DynaMight (top) and as pre-sented in the e2gmm publication (bottom).

Figure 2: Reconstruction of EMPIAR-10073 using DynaMight (top) and 3DFlex (bottom). Both maps are filtered and colored by local resolution estimated with cryosparc with an FSC threshold of 0.143.

Second, we also applied 3DFlex to our data set on the complete yeast inner kinetochore complex assembled onto the CENP-A nucleosome. We used default 3DFlex parameters, except the number of latent dimensions, which we set to the same number as we used in DynaMight (i.e. 5). The new Extended Data Figure 4 compares the improved reconstructions from DynaMight and 3DFlex. We observed an over-estimation of the local resolutions of 3DFlex, especially in regions where artefacts from the deformations are present. The reason for these inflated resolution estimates could be that the deformations are learnt with the same reference model and therefore are more consistent between the half-sets that are assigned after training.

The Results section on the spliceosome now includes these statements:

”Using this prior, the deformations estimated by DynaMight are qualitatively similar to those observed for the same data set using e2gmm (Extended Data Figure 1, Extended Data Video 3). For a different spliceosome data set (EMPIAR-10073), using the less informative, smoothness prior in DynaMight led to an improved reconstruction with better map features and higher local-resolution estimates (Extended Data Figure 2, Extended Data Video 4) than reported for 3DFlex, even though 3D classification in RELION-5 selected a structurally heterogeneous subset of only 86,624 particles, compared to 102,500 particles used for 3DFlex.”

The Results section on the kinetochore now includes these statements:

”Nevertheless, the DynaMight map had better protein and nucleic acid features than a map obtained for the same dataset with 3DFlex, using default parameters (Extended Data Figure 4). The DynaMight map also correlated better than the map from 3DFlex with atomic models that were built in the maps from multi-body refinement. Despite these observations, resolution estimates calculated from half-maps calculated by 3DFlex were higher than those calculated from half-maps by DynaMight. It is possible that the use of a single model of 3D deformations in 3DFlex,

as opposed to the refinement of two independent models in DynaMight, may lead to over-estimation of local resolution.”

2. Analyzing synthetic data could be highly beneficial. Other approaches have used synthetic data to demonstrate their potential best-case outcomes and limitations. Since real-world data lacks a concrete ground truth, synthetic data can provide significant insights.

Internally, we used a simulated data set of the SARS-Cov-2 spike protein to test the correctness of DynaMight. The dataset consists of 90,710 particle images generated from 5 states of an MD simulation. We processed the dataset with DynaMight using 15k Gaussian basis functions and 10 latent dimensions. With this setting we were able to recover all the distinct states with few false classifications (see Figure 3 below). We also reconstructed using the DynaMight backprojection and obtained a map with improved density in the flexible regions.

Figure 3: **A**: UMAP of the 10 dimensional latent space colored by the ground truth state corresponding to the particle images. The maps in the corresponding colors show a the Gaussian model that is predicted at these locations. **B**: The top image shows the PDBs of the 5 states and the bottom image shows the 5 Gaussian model maps of these states. **C**: Reconstruction using DynaMight left and consensus refinement right. **D**:

3. The manuscript quickly addresses compositional heterogeneity, stating that DynaMight cannot handle it and that it must be managed separately beforehand. However, removing this heterogeneity in biological samples is often challenging. I suggest a discussion on the consequences of running DynaMight on samples with compositional heterogeneity. We observed that if severe compositional heterogeneity is present in the data the decoder tries to learn physically implausible deformations, for example whole domains moving away if they are missing for some particles.

Whereas the treatment of continuous structural heterogeneity remains an active area of research in cryo-EM image processing, a wide variety of (relatively mature) tools exist for the separation of compositional heterogeneity. We chose not to allow Gaussians to appear or disappear, like they can in e2gmm, because molecular movements could be described by disappearance of Gaussians in one place and appearance of Gaussians in another. Moreover, allowing them to (dis)appear would further increase the number of parameters, and hence the scope for overfitting. We have included the following statements to the Discussion to make this reasoning more explicit:

"To avoid deformations to be described by the disappearance of Gaussians in one place and the appearance of Gaussians in another, and to limit the number of model parameters, DynaMight does not refine an occupancy factor for each Gaussian. Consequently, DynaMight cannot model compositional heterogeneity and it is unclear how it will perform on data sets, and such heterogeneity should be removed using existing discrete classification methods prior to the application of DynaMight."

4. The decoder's architecture is not as clearly described as the encoder's. Details about the training process, such as whether the system uses pre-trained weights or initializes them for each run, would be helpful. A comprehensive methods section detailing the implementation would be beneficial.

We have added more detailed descriptions of the method, including the decoder, to the new Extended Methods online section of the paper.

5. The authors caution against using model regularization, but include it as an option in the software as far as I understand the text. However, I don't see it in the GUI as an option. Is it retained? If it is: Since there's no absolute ground truth in this field, and it's difficult to assess the extent of model bias, the rationale for retaining this potentially problematic option should be clarified. Perhaps comparing its use with less biased techniques, like multi-body refinement, could offer more clarity. If it isn't: just clarify the text.

To prevent users from using the dangerous option of providing strong restraints derived from atomic

models, we have not exposed this option on the GUI. Instead, the RELION- 5 GUI only provides access to the relatively safe option of enforcing smoothness on the deformations without the use of an atomic model. All results reported in the paper We now explicitly mention this in the Implementation details of the Approach section, which now reads:

”Because, as we will show below, the direct regularisation of the deformation fields using atomic models may lead to overfitting, only the approach that enforces smoothness on the deformations, without the use of an atomic model, is exposed to the user on the GUI.”

Minor comment:

6. The visualization of the model fit in the density shown in Figure 3 could be improved. Its current form is not very informative, and alternative ways of representation might be more effective.

We have added Extended Data Videos 1 and 2 to make these fits clearer.

Reviewer 2:

In this manuscript, the authors presented a novel protocol that combines the Gaussian representation of protein heterogeneity analysis (similar to Chen 2021), and a deformation field that utilize the learned heterogeneity information to improve the quality of single particle reconstruction (similar to Herreros 2023). A VAE is used to learn the heterogeneity from the particles, and a separate neural network is used to bridge the Gaussian and the Cartesian grid representation and generate the deformation field. While the idea is conceptually attractive, the manuscript seems somewhat unfinished, with many details of the method implementation and the examples left out. This makes it difficult to judge the design of the method and the performance of the protocol.

We consider the manuscript to be finished, but it does describe an area of research that is not! In our opinion, explicitly pointing out the potential pitfalls of our method (and by extension alternative methods in the field that did not explore these) is an important contribution of this paper, even if the optimal solutions to some of these problems have yet to be developed. Nevertheless, in our revised manuscript, we show that DynaMight already operates at or beyond the current state-of-the-art in the field.

My major comments are listed below.

Many details on the method implementation and the processing of the experimental datasets are missing. For example, what is the loss function used during training? Is the resolution of the reconstruction used to regularize the training and prevent overfitting? How are the weights of regularization determined? The encoder is described to be a 3-layer fully connected network, but the structure of the decoder is not presented. In the examples shown in the paper, it is unclear how many Gaussian functions were used to represent the molecule, how well does the Gaussian representation fit the initial reconstruction, or what is the scale of the movement (how many Å a domain moves or degree it tilts). It would be good to have a Method section and supplementary materials to describe the details, otherwise it is hard to tell what are the differences from the existing methods and what are the improvements.

We have added an Extended Methods online section that describes the training details. The loss function used in training consists of two parts, a data-fidelity term (Eq 11) and a regularization term (Eqs 13 and 14). For the data term the L^2 norm between the generated image and the experimental image is used and the regularisation term penalizes distance changes of neighbouring Gaussians, as well as Gaussians being too close to each other. The data and regularisation term are weighted such that the ratio between the gradient coming from both terms is a user-defined number. We used 0.9 in all our experiments, meaning that the gradient from the data is slightly larger than the gradient of the regularisation functional. The resolution of the reconstruction is not used in the regularization. We now present the structure of the coordinate-based decoder in more detail in 'The Variational Autoencoder' subsection of the Extended Methods online section. We also added Extended Data Table 1 with details about the processing and the results for all four data sets.

The authors show potential model bias when information from atomic models were used for the analysis, but it is unclear what sort of constraints were applied. In Figure 2d, it seems that the atomic model was used to build the Gaussian model, and the distances between the bonded atoms were used to regularize the movement. How exactly is the Gaussian model initialized? Is the atomic model simply rigid-body fitted into the consensus map, or some sort of flexible fitting or Gaussian model optimization was performed? Is there any precaution taken for the gold-standard particle splitting, or the same atomic model was used for the analysis of the two half sets? How are the bond constraints applied? Is it determined from one existing model, or the ideal bonds length and angle from PDB statistics? How is the weighting between map and model regularization determined? For Figure 2b-c, it is mentioned that the bond constraints were not used, but how were the Gaussian model initialized in either case? If they were initialized the same way as Figure 2d, then obviously the same model bias gets into the system. If seeded directly from the volume, then it is hard to argue whether the difference comes from the Gaussian placement or the bond constraints. Overall, there is too little detail in the manuscript to

decipher what is actually happening.

We added a detailed explanation of the initialisation procedures and the restraints that were used for the different regularization scenarios in the 'Initialisation of the reference model' and the 'Loss functions and regularization' subsections of the Extended Methods online section. Not using any regularization on the Gaussians (Figure 2b) does not lead to map improvements. This option is therefore not recommended and it is inaccessible from the RELION GUI. When using atomic models for regularization (Figure 2d), we performed rigid-body fitting of the atomic models into the consensus reconstruction, and initialised a Gaussian reference model by coarse-graining; using one Gaussian per 3-4 atoms. In this scenario, we only optimized a 'global' width and amplitude parameter for the Gaussian model, and we used the same initial model for both half-sets. The drawbacks of this approach are described in the main text. As we do not recommend this option to our users, also this option is inaccessible from the RELION GUI. When not using an atomic model for regularization (Figure 2c), the Gaussian models are initialised by randomly filling the subvolume in the consensus map above a given density threshold. Different instances of this random procedure are used to initialise two different models for the two independent half-sets. This is the recommended procedure to run DynaMight and it is the one that is accessible from the RELION GUI. In none of the regularization scenarios is the map itself explicitly regularized, although some implicit regularization exists through its representation as Gaussian model. When regularization of the Gaussians is performed, the regularization and the data terms are weighted by their relative gradients (see above).

In the last example, the authors showed that the deformation based reconstruction is not performing as well as the multi-body refinement. Then in discussion, the authors argued there is a minimum size requirement for multi-body refinement, but not in the deformation based approach, so that the proposed method is still useful. However, none of those is shown experimentally. It is unclear what is the minimum size requirement for multi-body refinement, and there is no evidence that dynamight does not have that disadvantage. To truly show the new method is useful, it would be critical to have examples that multi-body does not work but the new method does, and discuss what is causing the differences.

The reviewer is probably correct in that we cannot say that DynaMight lacks a minimum size requirement. Therefore, we have removed this statement from the revised manuscript. We hope that our explicit comparisons with e2gmm and 3DFlex, as introduced in response to reviewer 1, convince the reviewer that (despite our observations that further improvements are probably still possible) DynaMight already operates at or beyond the current state-of-the-art in the cryo-EM field and will thus be useful.

Some less major comments:

For the encoder, it seems that the authors decided to use a fully connected network for real space images. This is quite unexpected, since convolutional networks for images have become

the mainstream for many years now. Is there any specific concern against convolutional networks? Usage of fully connected networks might cause overfitting, since it is hard to imagine it capturing translation, rotation and conformation of each particle from the image. One way to check how much overfitting the encoder-decoder introduces is to input translated/rotated copies of the same particle and see if it gets mapped to the same position in the latent space. Including a separate test dataset during training may help too.

We agree that this is somewhat unexpected. As now described in 'The Variational Au-toencoder' subsection of the Extended Methods online, we tried alternative architectures and also substituted the particle images with random vectors, which did not decrease performance by much, suggesting that the encoder is not picking up much information from the images and is not generalizing well. It is therefore possible that the encoder is not needed at all, and the latent representation could be optimized directly, similar to the approach in 3DFlex. Because the encoder is not a computational bottleneck, we decided to not change this. In ongoing work we are trying to design encoders that are better at extracting information from the images and that are better at generalization.

The authors mentioned the half-set splitting approach for heterogeneity validation, but in all three examples, only one latent space and conformations were shown (figure 5a for example). Are the conformations from one of the half sets? Are there any differences in the movement modes from the other half set? Is there a way to “align” the latent space from one VAE on one half set to another VAE’s output?

The examples in the paper show only the latent space and corresponding conformations from one half-set. We have added remarks to make this clearer to the legends of Figures 4 and 5. The overall organization of the latent spaces and the corresponding deformations tend to be similar for the two half-sets, although the latent spaces are typically not “aligned” (and we do not currently have methods to align them). We now illustrate these observations for the pre-catalytic spliceosome data set in Extended Data Figure 1.

There is no clear indication of where the datasets used in the second and third examples come from. Are they open datasets from EMPIAR, or private datasets that may only be available upon request? In data availability, it only mentioned there is no restriction but not where to find the data.

We have submitted these data sets to the EMPIAR data base and have added the corresponding accession numbers to the Data Availability section. In addition, we have uploaded all our improved reconstructions, plus their half-maps, to the EMDB.

Decision Letter, first revision:

Dear Sjors,

Thank you for submitting your revised manuscript "DynaMight: estimating molecular motions with improved reconstruction from cryo-EM images" (NMETH-A54182A). It has now been seen by the original referees and their comments are below (these are the same as the comments I shared with you for your response). I sent your response to these comments to Reviewer #2, and they are satisfied with the revision plan. However, while they (and we) are okay with not adding additional comparisons, we recommend citing the papers they mention and briefly discussing them in either the introduction or the discussion section of your paper. We are happy in principle to publish this paper in Nature Methods, pending minor revisions to satisfy the referees' final requests and to comply with our editorial and formatting guidelines.

TRANSPARENT PEER REVIEW

Nature Methods offers a transparent peer review option for new original research manuscripts submitted from 17th February 2021. We encourage increased transparency in peer review by publishing the reviewer comments, author rebuttal letters and editorial decision letters if the authors agree. Such peer review material is made available as a supplementary peer review file. Please state in the cover letter 'I wish to participate in transparent peer review' if you want to opt in, or 'I do not wish to participate in transparent peer review' if you don't. Failure to state your preference will result in delays in accepting your manuscript for publication.

Please note: we allow redactions to authors' rebuttal and reviewer comments in the interest of confidentiality. If you are concerned about the release of confidential data, please let us know specifically what information you would like to have removed. Please note that we cannot incorporate redactions for any other reasons. Reviewer names will be published in the peer review files if the reviewer signed the comments to authors, or if reviewers explicitly agree to release their name. For more information, please refer to our FAQ page.

ORCID

IMPORTANT: Non-corresponding authors do not have to link their ORCIDs but are encouraged to do so. Please note that it will not be possible to add/modify ORCIDs at proof. Thus, please let your co-authors

know that if they wish to have their ORCID added to the paper they must follow the procedure described in the following link prior to acceptance:

Sincerely,
Arunima

Arunima Singh, Ph.D.
Senior Editor
Nature Methods

Reviewer #1 (Remarks to the Author):

The authors have addressed all my concerns. I am especially happy that they put effort in comparing their approach fairly with other software. This is essential for the practitioner. Additionally, the software further improved significantly. I, thus, wholeheartedly recommend publication.

Reviewer #1 (Remarks on code availability):

I tested the software and it works as intended. Though installation especially on a cluster environment is unfortunately not trivial.

Reviewer #2 (Remarks to the Author):

In the revised version, the authors include a Method section, so I finally get to have a peek at what is inside the model, although I am still quite confused by some of the implementations. Here are my comments.

1. Some questions in my initial comments were not answered. I am listing them again here.

“In the examples shown in the paper, it is unclear how many Gaussian functions were used to represent the molecule”

- This is now explained when the coarse grain atomic structures are used, but it is still unclear how the number of Gaussian functions is decided when an atomic model is not used.

“how well does the Gaussian representation fit the initial reconstruction”

- It would be good to show a map-model FSC plot so we know to what extent the Gaussian model can be trusted.

“what is the scale of the movement (how many Å a domain moves or degree it tilts).”

- When the scale of movement is small, the problem can be solved as amplitude change (or by linear models like 3DVA), so it is good to have a quantitative measure.

2. If the authors were to make comparisons between methods, it would be good to compare to the more recent development, such as (Herreros 2023) and (Chen 2024). Both show improvement of reconstruction quality after addressing the heterogeneity so it would be easier to compare quantitatively.

3. In the new Method section, the authors describe a framework in which Gaussians of multiple distinct widths can be used (N_c in equation 8), except it is never used or discussed later. What is the intention of this design, and are there any cases that it might be helpful?

4. The description of the decoder structure is still confusing. The authors cited the attention based design in [32], but it is not clear how it is implemented here. How are the spatial encoding input into the model? Does the spatial position encoding actually improve the results? Is there a separate branch of the neural network for the spatial encoding input or it is simply stacked with the conformation input? It would be good to have a diagram for this.

5. The equations in decoder design are not very understandable. What exactly is the decoder output? A ($N_g \times 3$) vector per image? What is $D(z_i, x)$ in equation 8? Is it the deformation in 2D or 3D? How is it different from $(X_i)_i$ in equation 9? It would also be good to re-state the variables in the main text so readers don't have to search for the references.

6. It seems that the authors project the Gaussian functions in real space, then perform Fourier transform? It is unclear why since the Fourier transform of Gaussian functions is still Gaussian.

7. In the Training section, it is described “we replace the positions of the Gaussians of the reference model by the predicted Gaussian positions with the smallest displacement from the current reference model. The latter ensures that the reference model is in the distribution of deformed models.” Does the reference model ever move out of the distribution without this regularization? Is there really any disadvantage when it happens?

Reviewer #2 (Remarks on code availability):

The organization and documentation of the code seem reasonable. I did not actually install the program, because pip tends to pollute the python environment...

Author Rebuttal, first revision:

In the revised version, the authors include a Method section, so I finally get to have a peek at what is inside the model, although I am still quite confused by some of the implementations. Here are my comments.

We thank the reviewer for their careful analysis of our method. We note that they have not expressed any negative opinions about our manuscript. They merely ask for additional information, most of which we provide in a point-by-point response below.

1. Some questions in my initial comments were not answered. I am listing them again here.

"In the examples shown in the paper, it is unclear how many Gaussian functions were used to represent the molecule"

- This is now explained when the coarse grain atomic structures are used, but it is still unclear how the number of Gaussian functions is decided when an atomic model is not used.

We recommend using 2 Gaussians per residue as a rule of thumb. Nevertheless, DynaMight can be run with an arbitrary number of Gaussians. If computational resources are limited or a low resolution estimation of the motion is desired a lower number can be used. We will add a statement to this effect to the final version of the paper.

"how well does the Gaussian representation fit the initial reconstruction"

- It would be good to show a map-model FSC plot so we know to what extent the Gaussian model can be trusted.

We suggest to include the below figure in the Extended Data of the final version of our manuscript.

Figure 1: Fourier shell correlation (FSC) between the Gaussian consensus model, of half set 1, and the reconstructed map from the consensus refinement (blue) and between the final Gaussian model and the improved DynaMight reconstruction (red). The dashed cyan line indicates the reported resolution of the masked consensus refinement.

"what is the scale of the movement (how many Å a domain moves or degree it tilts)."

- When the scale of movement is small, the problem can be solved as amplitude change (or by linear models like 3DVA), so it is good to have a quantitative measure.

We provide the maximal and mean RMSD of the Gaussian model within the whole dataset in Extended Data Table 1. For the reviewer's convenience, we copy the table below.

	Spliceosome	tri-snRNP	CCAN:CENP-A	Kinetochores
EMPIAR	10180	10073	11910	11890
EMDB-id	EMD-19791	EMD-19789	EMD-19799	EMD-19794
nr. particles	44,537	86,624	100,311	108,672
box size (pixel)	320	380	360	320
pixel size (Å)	1.699	1.4	0.853	1.08
nr. gaussians	30,000	30,000	30,000	30,000
nr. epochs	221	273	373	290
nr. latent dimensions	5	10	10	10
compute time (h)	9+6+4	24+7+16	17+8+20	27+11+24
GPU used	A6000	A6000	A100	A100
mean deformations (Å)	7.755	4.303	3.608	4.112
max deformation (Å)	24.139	14.071	8.248	8.802

2. If the authors were to make comparisons between methods, it would be good to compare to the more recent development, such as (Herreros 2023) and (Chen 2024). Both show improvement of reconstruction quality after addressing the heterogeneity so it would be easier to compare quantitatively.

The reviewer asked for a more detailed description of our method in the first round of review, leading to additional questions after a second round of review, which we now address in this document. However, the request for comparisons with the approaches by Herreros and by Chen is a new one. It was not raised in the original review and it is not related to the additional methodological details provided after the first round of review. We therefore prefer not to engage with this request at this stage.

We further note that we had already put considerable work into comparisons between DynaMight and Flex3D and the original approach by Chen for our previous revision, and that reviewer #1 is *“especially happy that they put effort in comparing their approach fairly with other software”*.

3. In the new Method section, the authors describe a framework in which Gaussians of multiple distinct widths can be used (N_c in equation 8), except it is never used or discussed later. What is the intention of this design, and are there any cases that it might be helpful?

This observation is correct. We included the description of N_c for completeness, since it reflects the implementation. Although it was not used in the results of the paper, it could be helpful if the consensus map contains large variation in local resolution and the same width of all Gaussians does not give a reasonable representation of the map. We will add a statement to explain this in the final version of the manuscript.

4. The description of the decoder structure is still confusing. The authors cited the attention based design in [32], but it is not clear how it is implemented here. How are the spatial encoding input into the model? Does the spatial position encoding actually improve the results? Is there a separate branch of the neural network for the spatial encoding input or it is simply stacked with the conformation input? It would be good to have a diagram for this.

Instead of concatenating the 3 dimensional coordinates to the latent representation, we enhance them by a fixed positional encoding. We use the sine and cosine function for lifting the 3-dimensional position to a higher dimensional space similar to equation (4) in [MILDENHALL, Ben, et al. Nerf: Representing scenes as neural radiance fields for view synthesis. *Communications of the ACM*, 2021, 65. Jg., Nr. 1, S. 99-106]. We will add this reference to the final version of the paper. Then, the input to the multi-layer perceptron is the concatenation of the positional encoding of a single 3D position concatenated with the conformation input (See Figure 1, which we will add as an Extended Data Figure to the final version of the paper). During training, the decoder network is queried for every position of the consensus Gaussian model.

We observed that without the positional encoding the deformations are too smooth and that localized motion is not captured well. We will add a statement to explain this in a final version of the manuscript.

Figure 1: Diagram of the decoder architecture. The queried position is lifted to a higher dimensional space

via a fixed positional encoding function, where the lifting dimension is defined by N_p . The N_l dimensional latent code is concatenated with the encoded position, and input to a multilayer perceptron (MLP), which outputs a 3 dimensional displacement vector. To obtain the final position the displacement vector is added to the original position x .

5. The equations in decoder design are not very understandable. What exactly is the decoder output? A $(N_g \times 3)$ vector per image? What is $D(z_i, x)$ in equation 8? Is it the deformation in 2D or 3D? How is it different from $(X_i)_i$ in equation 9? It would also be good to re-state the variables in the main text so readers don't have to search for the references.

Per image, the decoder indeed outputs a $(N_g \times 3)$ vector describing the displacements of the Gaussians in the consensus model. Equation 8 is just a general formula of describing the density by a Gaussian model, without modeling the deformation. When modeling the deformation x in Equation 8 is substituted by $D(z_i, x)$. The deformation is modeled in 3D and $D(z_i, x)$ is a 3 dimensional vector.

X_i in equation 10 denotes the projected 3D positions, i.e. $X_i^j = P_i(D(z_i, c_j))$, where P_i is the projection operator corresponding to the i -th image and c_j is the 3D position of the j -th Gaussian in the consensus model.

In the final version of the paper, we will make sure that all variables are defined close to where they are used.

6. It seems that the authors project the Gaussian functions in real space, then perform Fourier transform? It is unclear why since the Fourier transform of Gaussian functions is still Gaussian.

For numerical efficiency we chose to implement the image formation in real space. In doing so, we don't have to compute the sum over all Gaussians that would be necessary in Fourier space, but instead deal only with a few pixel values per Gaussian in real-space. Although in Chen 2023 (Integrating Molecular Models Into CryoEM Heterogeneity Analysis Using Scalable High-resolution Deep Gaussian Mixture Models) an alternative method for efficient computation based on the separability of Gaussians was developed, we decided that a real-space implementation suited our case better.

7. In the Training section, it is described "we replace the positions of the Gaussians of the reference model by the predicted Gaussian positions with the smallest displacement from the current reference model. The latter ensures that the reference model is in the distribution of deformed models." Does the reference model ever move out of the distribution without this regularization? Is there really any disadvantage when it happens?

We observed that the reference model can move out of distribution, sometimes even to a point where the structure is completely distorted. As long as the deformations satisfy the regularization constraints this would not change the value of the loss function, but we observed that this can lead to unphysical displacements of the Gaussians and suboptimal reconstructions. We will add a statement to explain this in the final version of the manuscript.

Final Decision Letter:

Dear Sjors,

I am pleased to inform you that your Article, "DynaMight: estimating molecular motions with improved reconstruction from cryo-EM images", has now been accepted for publication in Nature Methods. The received and accepted dates will be October 18, 2023 and July 3, 2024. This note is intended to let you know what to expect from us over the next month or so, and to let you know where to address any further questions.

Over the next few weeks, your paper will be copyedited to ensure that it conforms to Nature Methods style. Once your paper is typeset, you will receive an email with a link to choose the appropriate publishing options for your paper and our Author Services team will be in touch regarding any additional information that may be required. It is extremely important that you let us know now whether you will be difficult to contact over the next month. If this is the case, we ask that you send us the contact information (email, phone and fax) of someone who will be able to check the proofs and deal with any last-minute problems.

Please note that *Nature Methods* is a Transformative Journal (TJ). Authors may publish their research with us through the traditional subscription access route or make their paper immediately open access through payment of an article-processing charge (APC). Authors will not be required to make a final decision about access to their article until it has been accepted. Find out more about Transformative Journals

If you are active on Twitter/X, please e-mail me your and your coauthors' handles so that we may tag you when the paper is published.

Best regards,
Arunima

Arunima Singh, Ph.D.
Senior Editor
Nature Methods